# Visualizing the functional architecture of the endocytic machinery

**Andrea Picco, Markus Mund, Jonas Ries, François Nédélec, Marko Kaksonen\***

Cell Biology and Biophysics Unit, European Molecular Biology Laboratory, Heidelberg, Germany

**Abstract** Clathrin-mediated endocytosis is an essential process that forms vesicles from the plasma membrane. Although most of the protein components of the endocytic protein machinery have been thoroughly characterized, their organization at the endocytic site is poorly understood. We developed a fluorescence microscopy method to track the average positions of yeast endocytic proteins in relation to each other with a time precision below 1 s and with a spatial precision of ∼10 nm. With these data, integrated with shapes of endocytic membrane intermediates and with superresolution imaging, we could visualize the dynamic architecture of the endocytic machinery. We showed how different coat proteins are distributed within the coat structure and how the assembly dynamics of N-BAR proteins relate to membrane shape changes. Moreover, we found that the region of actin polymerization is located at the base of the endocytic invagination, with the growing ends of filaments pointing toward the plasma membrane.

**\*For correspondence:** kaksonen@embl.de

**Competing interests:** The authors declare that no competing interests exist.

**Reviewing editor**: Suzanne R Pfeffer, Stanford University, United States

## Introduction

Clathrin-mediated endocytosis is a key membrane trafficking process for the uptake of cargo molecules from the cell surface. It is involved in numerous different biological contexts from the generation of cellular polarization, to virus uptake and the regulation of neural signaling. The formation of clathrin-coated endocytic vesicles is driven by a complex molecular machinery composed of more than 50 different proteins (*Doherty and McMahon, 2009*; *Boettner et al., 2012*; *Weinberg and Drubin, 2012*). These proteins recruit cargo molecules and reshape the plasma membrane to generate the endocytic vesicle. The assembly dynamics of the endocytic protein machinery has been described in detail by live-cell imaging (*Gaidarov et al., 1999*; *Merrifield et al., 2002*; *Kaksonen et al., 2003*, *2005*; *Newpher et al., 2005*; *Sirotkin et al., 2010*; *Loerke et al., 2011*; *Taylor et al., 2011*; *Cocucci et al., 2012*; *Berro and Pollard, 2014*). However, the spatial organization of the endocytic proteins during vesicle budding remains poorly understood. The complexity and the size of the endocytic machinery, below the resolution limit of light microscopy, have made it challenging to reveal how the endocytic proteins are organized during vesicle budding.

Budding yeast *Saccharomyces cerevisiae* has been used extensively as a model organism to study endocytosis. The protein machinery of clathrin-mediated endocytosis is largely conserved between mammals and yeast (*Conibear, 2010*). In yeast, endocytosis starts with the recruitment of clathrin and several clathrin adaptors and accessory proteins to the plasma membrane to initiate the assembly of the vesicle coat (*Carroll et al., 2012*; *Godlee and Kaksonen, 2013*). During this initial early phase, which can last between 40 and 90 s, the coat assembles on a flat plasma membrane and cargo molecules are recruited to the endocytic site. Later, additional coat proteins and actin interactors are recruited. The actin interactors include Las17 (homolog of mammalian N-WASP) and type I myosins Myo3 and Myo5, which both activate the actin filament nucleating Arp2/3 complex (*Winter et al., 1999*; *Evangelista et al., 2000*; *Lechler et al., 2000*). In yeast, the invagination of the plasma membrane, and the consequent inward movement of the coat, is concomitant with the appearance of

**eLife digest** Cells take up proteins and other useful material (called cargo) from their external environment through a process known as endocytosis. To start with, the cargo accumulates in a patch on the surface of the cell. On the inner side of the cell's membrane, a protein called clathrin gathers around the patch of cargo. Clathrin molecules and many other proteins bind together to make a lattice-like coat that causes the membrane to curve inwards and form a pocket that contains the cargo. This continues until the cargo is completely surrounded by membrane and eventually forms a bubble-like structure, or 'vesicle', that moves into the cell.

More than 50 other proteins are involved in the endocytosis. These proteins arrive at the site of endocytosis in a particular order, complete their tasks and then move away to be used in further rounds of endocytosis. It is not clear how these proteins are organized to complete these steps because it is technically difficult to track the movements of many proteins at the same time.

Here, Picco et al. developed a new fluorescence microscopy method that enabled them to track the positions of many of the proteins involved in endocytosis in yeast cells in real time. The experiments revealed when the proteins arrived at the site of endocytosis and how they assembled in relation to the membrane. For example, a group of proteins called N-BAR proteins formed an extended lattice covering the sides of the pocket that forms as the membrane curves inwards.

To transform the flat membrane into a vesicle, a network of filaments made of a protein called actin needs to form at the site of endocytosis. The new method shows that the actin filaments grow in a small region at the base of the developing vesicle. By combining different types of microscopy data, Picco et al. were able to build a comprehensive model describing when the proteins involved in endocytosis move and assemble.

The next challenge will be to understand the physics behind the molecular machine composed of these many proteins and the cell membrane.

actin at the endocytic site (*Kukulski et al., 2012*). A functional dependence on actin polymerization is possibly due to high turgor pressure (*Aghamohammadzadeh and Ayscough, 2009*). In mammals, actin is similarly critical under conditions of high membrane tension (*Boulant et al., 2011*). Two of the coat-associated proteins, Sla2 (homolog of mammalian Hip1R) and the epsin Ent1, interact directly with lipids, clathrin and actin, and are essential for mediating the forces from the actin cytoskeleton to deform the membrane (*Baggett et al., 2003*; *Sun et al., 2005*; *Skruzny et al., 2012*). The shape of the endocytic invagination is regulated by proteins containing BAR domains, such as the Rvs161/167 heterodimer, which are localized at the neck region of the membrane invagination (*Idrissi et al., 2008*; *Youn et al., 2010*; *Kishimoto et al., 2011*). After vesicle scission the endocytic machinery quickly disassembles and the free vesicle is trafficked further into the cell.

The exact molecular mechanisms of endocytosis remain unknown, largely due to lack of knowledge about the functional organization of the protein components of the endocytic machinery. Here, we developed a novel imaging approach to characterize the spatial and temporal relationship between the different protein components of the endocytic machinery during vesicle budding. Furthermore, we integrated this data with time-resolved membrane shapes that were obtained previously by correlated light and electron microscopy (*Kukulski et al., 2012*). With this approach, we determined several key features of the dynamic architecture of the endocytic machinery during vesicle budding (*Figure 1A*).

## Results

### High-precision tracking of endocytic protein dynamics in living cells

To understand how the endocytic machinery is organized during vesicle budding, we developed an imaging approach to track the movements of different fluorescently labeled endocytic proteins in relation to each other in living cells. We used a two-step procedure: in the first step, we analyzed different endocytic proteins individually to measure their average dynamic behavior and abundance. In the second step, we aligned the resulting data from these different proteins relative to each other in space and time using simultaneous two color imaging. This two-step procedure allowed us to separately optimize the acquisition rate and the alignment precision for single and double channel imaging respectively.

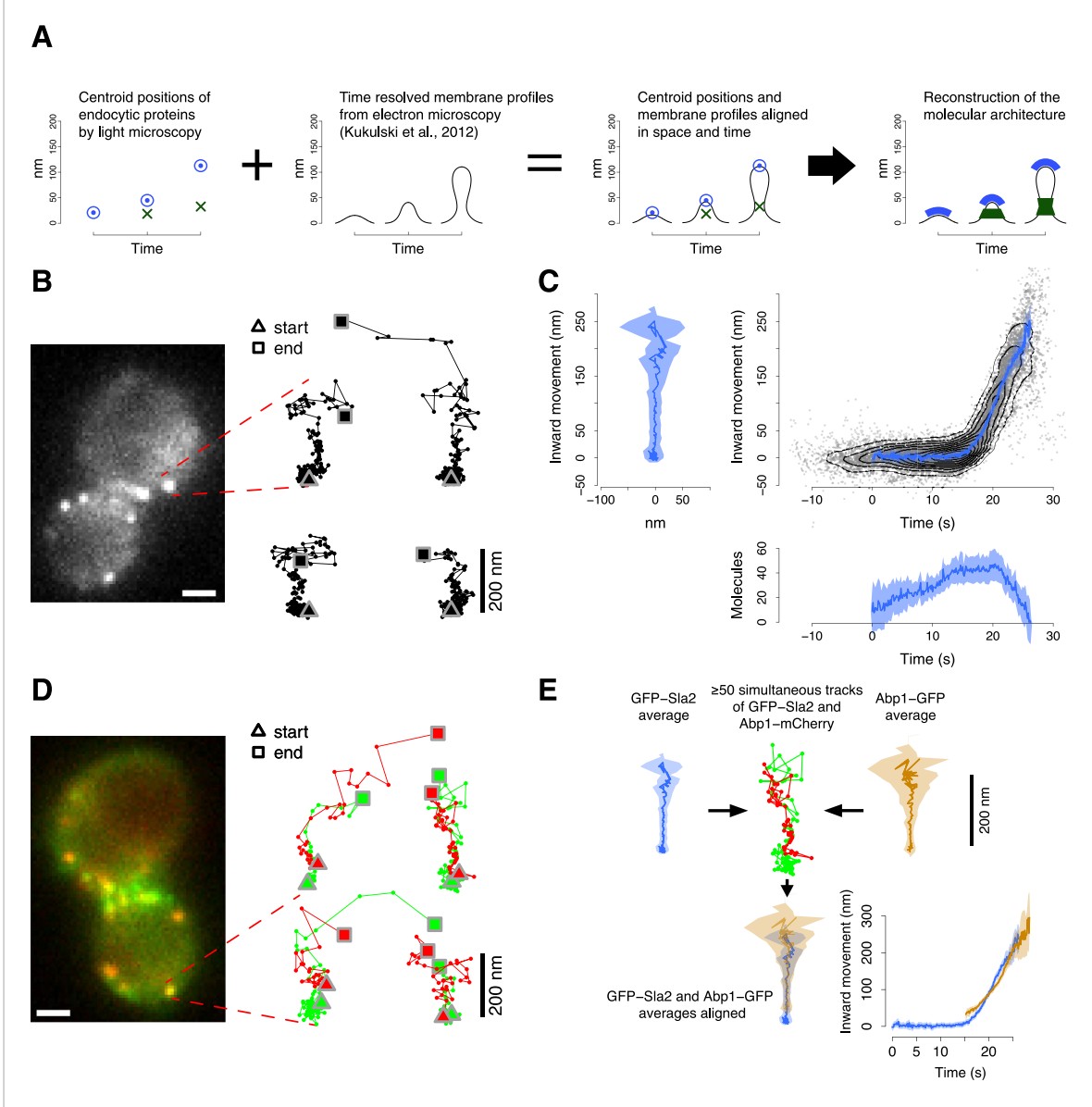

**Figure 1**. Tracking procedure. (**A**) The rational behind our approach. The centroid positions of endocytic proteins were correlated with the plasma membrane intermediates derived from CLEM (*Kukulski et al., 2012*). We could thus position the protein complexes along the plasma membrane invagination to reconstruct the molecular architecture of the endocytic machinery. (**B**) A yeast cell expressing the coat protein Sla2 tagged N-terminally (GFP-Sla2) and a collection of trajectories of GFP-Sla2 centroid in different endocytic events. The trajectories are oriented so that the plasma membrane lies horizontally at the bottom and the inward movement axis is vertical. The triangle and the square mark the start and the end of each trajectory, respectively. (**C**) The information content of the GFP-Sla2 average trajectory: The average trajectory movement on the focal plane. The trajectory is aligned so that the Y-axis represents the inward movement along the invagination and the X-axis represents the movement along the plasma membrane (left panel). The inward movement of the trajectories over time (right panel). The number of molecules recruited at the endocytic site over time (bottom panel). The 53 individual trajectories that were used to generate this average are plotted (gray) together with the average (blue). The contour lines highlight point densities. (**D**) A yeast cell expressing GFP-Sla2 and the reference protein Abp1-mCherry and a collection of trajectories derived from the simultaneous acquisition of the target and reference proteins. (**E**) A diagram summarizing the steps that led to the spatial and temporal alignment of the average trajectories. Abp1 is used as the spatial and temporal reference. The average trajectory of the target protein (GFP-Sla2 in this example) is aligned to the average trajectory of Abp1-GFP (the reference protein) by aligning each of them to the respective trajectories acquired simultaneously in cells expressing GFP-Sla2 and Abp1-mCherry. More than 50 endocytic events are used to derive the average spatial and temporal transformation that aligns the average trajectories together. The

*Figure 1. continued on next page*

*Figure 1. Continued*

shading represents the confidence interval (See 'Materials and methods'). Scale bars in images are 1 µm long. See also *Figure 1—figure supplements 1,2,3,4,5*.

The following figure supplements are available for figure 1:

**Figure supplement 1**. Average trajectories of endocytic events.

**Figure supplement 2**. Experimental controls for the alignment procedure.

**Figure supplement 3**. Simulation of the accuracy of the two color alignment procedure.

**Figure supplement 4**. Simulation of the robustness to systematic shifts between the two channels during two color acquisition.

**Figure supplement 5**. Simulation of the accuracy of the trajectory averaging.

We used wide-field epifluorescence microscopy to image GFP-tagged endocytic proteins expressed from their endogenous genomic loci in haploid cells of yeast *S cerevisiae*. We acquired videos at the equatorial plane of the yeast cells, where the average direction of endocytic vesicle budding is planar with the focal plane (*Kaksonen et al., 2003*; *Kukulski et al., 2012*). We thus directly visualized the movements of the endocytic proteins along the axis of membrane invagination (*Kaksonen et al., 2003*; *Galletta et al., 2008*).

For each GFP-tagged target protein we tracked the centroid position and measured the fluorescence intensity during 50–80 individual endocytic events (*Figure 1B*, *Table 1*). In yeast, the dynamic behavior of the endocytic proteins during vesicle budding is highly stereotypical (*Kaksonen et al., 2003*; *Mooren et al., 2012*). We could therefore align the individual centroid trajectories in time and space to calculate average trajectories (*Figure 1C*, *Figure 1—figure supplement 1*). For this purpose we developed an algorithm that finds the translation, rotation and temporal shift minimizing the weighted mean square displacement between the time points of all pairs of trajectories, where the weights are the fluorescence intensities at each time point (see 'Materials and methods'). The centroid gives an estimate of the position of the center of mass of the protein distribution; therefore the

**Table 1**. Number of trajectories used in this study

|  | Number of single color trajectories to generate the average trajectory | Number of trajectory pairs used for spatial and temporal alignment |
| --- | --- | --- |
| GFP-Sla2 | 53 | 271 |
| Sla2-GFP | 55 | 158 |
| Sla1-GFP | 66 | 58 |
| End3-GFP | 75 | 57 |
| Abp1-GFP | 65 | NA |
| GFP-Act1 | 83 | 92 |
| Arc18-GFP | 69 | 90 |
| Rvs167-GFP | 58 | 340 |
| Las17-GFP | NA | 49 |
| Myo5-GFP | NA | 50 |
| TOTAL | 524 | 1165 |

The number of events that were tracked and used to compute the average trajectories and to align those average trajectories together.

average trajectory describes the stereotypic dynamic behavior of the center of mass of the target protein molecules during endocytosis.

We estimated the average number of molecules present at the endocytic site by calibrating the fluorescence signal using a kinetochore protein of known abundance (*Joglekar et al., 2006*; *Lawrimore et al., 2011*). For each protein, we used the average number of molecules to calibrate its fluorescence intensity profile with the absolute numbers of molecules (*Figure 1C*; see 'Materials and methods').

To control that the GFP-tagging did not impair the functionality of the tagged protein we monitored the lifetime of Abp1 patches, which is very sensitive to defects in the endocytic function (*Kaksonen et al., 2005*). We analyzed the lifetime of Abp1-mCherry patches, when Abp1-mCherry was coexpressed with each of the GFP-tagged proteins (see 'Materials and methods'). The Abp1-mCherry lifetimes were unaffected by all the GFP-tagged proteins except Las17-GFP, for which Abp1-mCherry lifetime was slightly longer (*Figure 1—figure supplement 2A*). Las17-GFP, however, must be mostly functional because it does not lead to the strong block of endocytosis observed with LAS17 deletion (*Sun et al., 2006*).

In the second step, we aligned the average trajectories of all proteins in space and time to a common reference. We chose Abp1 as a reference protein because of its abundance at the endocytic site, which leads to a strong fluorescence signal, and its highly regular dynamic behavior, which is directed perpendicular to the cell surface thereby defining the direction of the invagination movement (*Figure 1—figure supplement 2B,C*). We imaged strains coexpressing each GFP-tagged protein of interest together with Abp1-mCherry. The two colors were imaged simultaneously. Chromatic aberration was measured using multicolored fluorescent microbeads and corrected for. For each protein pair we recorded 50–350 endocytic events from which we extracted the centroid positions and fluorescence intensities of the two proteins as paired trajectories (*Figure 1D*, *Table 1*). From these paired trajectories we estimated the rotation, translation and time shift that optimally align the average trajectory of the protein of interest to the average trajectory of Abp1 (*Figure 1E*; see 'Materials and methods'). We thus produced a dataset in which all average trajectories are aligned to Abp1.

The axis of the invagination might not be perfectly aligned with the focal plane and would therefore be imaged as a projection, which leads to an underestimate of the true centroid movement along the invagination axis. Considering the distribution of the angles of Abp1 trajectories (*Figure 1—figure supplement 2C*), the spherical geometry of the cells and the depth of field (*Figure 1—figure supplement 2D*) we estimated that these projection effects would lead maximally to about 10% underestimation of the true movement along the invagination axis.

To measure the reproducibility of the alignment procedure we repeated multiple times the acquisition of paired trajectories for GFP-Sla2 and Rvs167-GFP expressed together with Abp1-mCherry. We also used paired trajectories of Abp1 derived from a strain expressing Abp1 tagged C-terminally with both mCherry and GFP (see 'Materials and methods'). The standard deviations in space and time of the repeated alignments were 2.8 nm and 0.1 s for Abp1, 5.5 nm and 0.3 s for Rvs167 and 12.9 nm and 0.6 s for Sla2 (*Figure 1—figure supplement 2E*). The accuracies of the alignments correlate with the signal intensities of these different proteins.

To further test the alignment procedure we created virtual 'ground truth trajectories' from which we generated sets of virtual trajectories by adding different levels of random noise. We first tested the accuracy of the two color alignment procedure using sets of virtual paired trajectories. With experimentally relevant noise levels the trajectories aligned within 3 nm from the ground truth trajectory (*Figure 1—figure supplement 3*).

To test the effect of misalignment between the color channels we took paired trajectories and shifted them artificially in relation to each other to different extents. Due to random orientations of the trajectories the color shifts average out and the increase in the shift is only manifested as increased uncertainty of track positions. With realistic color shifts (up to 50 nm) the average trajectories are only about 3 nm from the ground truth trajectory (*Figure 1—figure supplement 4*).

Finally, we tested the full two-step alignment procedure using virtual trajectories. This test demonstrated that the alignment procedure robustly generates representative average trajectories from noisy data (*Figure 1—figure supplement 5*).

In summary, with the average trajectories we can resolve the position and movement of different endocytic components in relation to each other along the direction of invagination with a temporal precision below 1 s and with a spatial precision of ~10 nm. In addition, we could complement the tracking data with time-resolved estimates of the numbers of molecules present at the endocytic site.

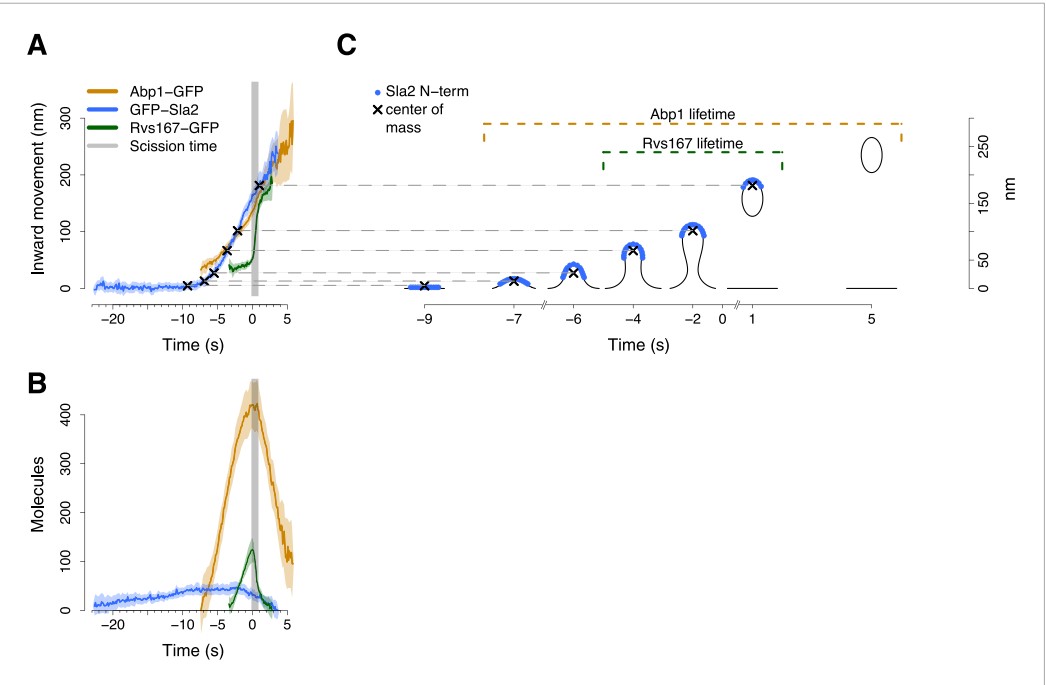

**Figure 2**. Alignment of trajectories and interpretation. (**A**) Abp1-GFP, GFP-Sla2 and Rvs167-GFP average trajectories aligned with each other. (**B**) The average number of molecules recruited at the endocytic locus varies from protein to protein, ranging from ~40 molecules at the peak of GFP-Sla2 to ~410 molecules at the peak of Abp1-GFP. (**C**) The lifetime of Abp1-GFP and Rvs167-GFP, as well as the centroid position of GFP-Sla2 are used to align in space and time the average plasma membrane profiles obtained by correlative light and electron microscopy (*Kukulski et al., 2012*). The centers of mass of the Sla2 model are marked by an 'X'. The grey vertical bar represents the estimated time window during which scission happens (*Kukulski et al., 2012*). The shading represents the confidence interval. The trajectories plotted are listed in *Supplementary file 1*.

## Correlating the tracking data with the changing membrane shape

The endocytic protein machinery progressively changes the shape of the plasma membrane, ultimately leading to vesicle budding. To provide a framework for interpreting the tracking data, we aligned the trajectories with time-resolved average membrane shapes obtained previously by CLEM (*Kukulski et al., 2012*), starting from a flat membrane and ending with a vesicle (*Figure 2*).

We used Sla2, Rvs167 and Abp1 proteins as spatial and temporal landmarks to align the membrane shapes from the CLEM study to our tracking data. These three proteins represent different endocytic substructures: the coat (Sla2), the invagination neck (Rvs167) and the actin cytoskeleton (Abp1) (*Kaksonen et al., 2005*; *Idrissi et al., 2008*). We generated aligned average trajectories and measured the average number of molecules for the proteins tagged N-terminally (GFP-Sla2) or C-terminally (Rvs167-GFP and Abp1-GFP; *Figure 2A,B*). We first defined a common frame by setting the initial position of the Sla2 centroid to coincide with the position of the flat plasma membrane. In GFP-Sla2 the GFP molecule is adjacent to Sla2's lipid-binding ANTH domain. The binding of Sla2's ANTH domain to PIP$_2$ at the plasma membrane is essential for productive endocytosis (*Sun et al., 2005*). We could therefore assume that before membrane bending starts (first time point in *Figure 2A,C*) the centroid of GFP-Sla2 is likely to be within few nm from the plasma membrane surface, whose position we took as the origin of the position axis. We then calculated the positions of the center of mass of Sla2 molecules on the shapes of the different membrane intermediates taking into account that Sla2 molecules cover a ~30–40 nm long region at the tip of the invagination (*Idrissi et al., 2012*). We used these positions to shift the membrane intermediates in time so that they coincided with the centroids in the average trajectory of GFP-Sla2 (*Figure 2A,C*).

Our alignment agreed well with the time resolved data from CLEM study: the start of Abp1 assembly coincides with initial membrane bending, and the assembly of Rvs167 starts when the invaginations are ~50 nm long (*Kukulski et al., 2012*) (*Figure 2C*). The CLEM data also indicated that

on average the vesicle scission takes place when ~59% of the Rvs167-GFP patch lifetime has passed (*Kukulski et al., 2012*). This time point coincides with the peak number of Rvs167-GFP molecules (*Figure 2B*). We therefore used the time at which the number of Rvs167-GFP molecules peaks as an estimate of the scission time and we defined it as the origin of the time axis (*Figure 2A,B*).

## Organization of coat associated proteins

To gain insights into the organization of the endocytic coat we focused on three coat-associated proteins: Sla2, Sla1 and End3. Sla2 is critical for membrane-actin coupling during vesicle budding (*Sun et al., 2005*; *Boettner et al., 2011*; *Skruzny et al., 2012*), whereas Sla1 and End3 are involved in the regulation of the initiation of actin polymerization (*Tang et al., 2000*; *Rodal et al., 2003*; *Kaksonen et al., 2005*).

Sla2 is composed of the N-terminal lipid-binding ANTH domain and a C-terminal actin-binding domain, which are separated by a long coiled-coil region. In Hip1R, the mammalian homolog of Sla2, the coiled-coil region has been measured to be ~40 nm long (*Engqvist-Goldstein et al., 2001*). We imaged cells expressing C-terminally GFP-tagged Sla2 (Sla2-GFP). Both GFP-Sla2 and Sla2-GFP average trajectories exhibited an initial motionless phase followed by inward movement. However, the Sla2-GFP trajectory was 29 ± 5 nm (mean ± SE) distant from the GFP-Sla2 trajectory during the initial non-motile phase (*Figure 3A*).

To confirm the spatial separation between GFP-Sla2 and Sla2-GFP we used a different imaging strategy based on double labeling: we tagged the Sla2 molecule at its N-terminus with GFP and at its C-terminus with mCherry (GFP-Sla2-mCherry; *Figure 3—figure supplement 1A*). We immobilized the Sla2 patches by treating the cells with latrunculin A, which inhibits actin polymerization and prevents membrane invagination at endocytic sites (*Kukulski et al., 2012*), and acquired still images of both GFP and mCherry signals at equatorial planes of the cells. The GFP-Sla2-mCherry molecules were oriented perpendicularly to the plasma membrane (*Figure 3—figure supplement 1B*) and the distance between the centroids of the GFP and mCherry tags was 33 ± 3.4 nm (mean ± SE) (*Figure 3—figure supplement 1C–F*, 'Materials and methods'). This compares well with the average distance between GFP-Sla2 and Sla2-GFP average trajectories during the initial motionless phase. These results indicate that Sla2 molecules are oriented within the coat so that the PIP$_2$ binding N-terminal domain is at the membrane and the actin-binding C-terminal domain projects into the cytoplasm.

When the membrane invagination starts, the GFP-Sla2 and Sla2-GFP centroids start moving inward and the distance between them becomes shorter (*Figure 3A*, *Figure 3—figure supplement 1G*). A simple explanation for this convergence is that the Sla2 molecules maintain a constant orientation with respect to the membrane. Therefore, during invagination the Sla2 molecules would reorient to accommodate membrane bending and the center of mass positions of their N and C-termini would move closer to each other (*Figure 3A*, right panel, time points −7, −6 and −4 s). Alternatively, a conformational change of Sla2 could also contribute to the convergence of the trajectories by bringing the N- and C-termini closer to each other.

We studied Sla1 and End3 proteins with C-terminal GFP-tags. Sla1 and End3 are multi-domain proteins but the locations of their N- and C-termini in the tertiary structure are less well characterized than in Sla2. The average trajectories of Sla1-GFP and End3-GFP had very similar shapes, but were separated by about ~10 nm or more (*Figure 3B* and *Figure 3—figure supplement 1G*).

The number of Sla2 molecules remained relatively constant at ~40 molecules during vesicle budding and started to drop ~2–3 s before scission (*Figure 3C*). Sla1 and End3 molecules peaked at ~90 and ~60 molecules respectively and started disassembling already during invagination (*Figure 3C*).

The overall dynamic behaviors of the Sla1 and End3 centroids appeared similar to that of Sla2. However, the Sla2 centroid started moving ~2 s before the Sla1 and End3 centroids and moved past the Sla1 centroid during invagination (*Figure 3B*, left panel). This suggests that Sla2, Sla1 and End3 are not distributed similarly within the coat. The observed trajectories could result from Sla2 being located on the invagination tip and Sla1 and End3 being located at the rim of the coat (*Figure 3B*, right panel). With such distributions, when the membrane starts bending, Sla2 molecules would start moving first, followed slightly later by Sla1 and End3 molecules.

The average trajectories describe the center of mass position of the tagged protein molecules, but the individual proteins could be distributed in various ways around the center of mass. We used superresolution microscopy (*Betzig et al., 2006*; *Hess et al., 2006*; *Rust et al., 2006*) to resolve the distribution of individual Sla1 proteins at the endocytic site: We fixed cells expressing Sla1 fused to

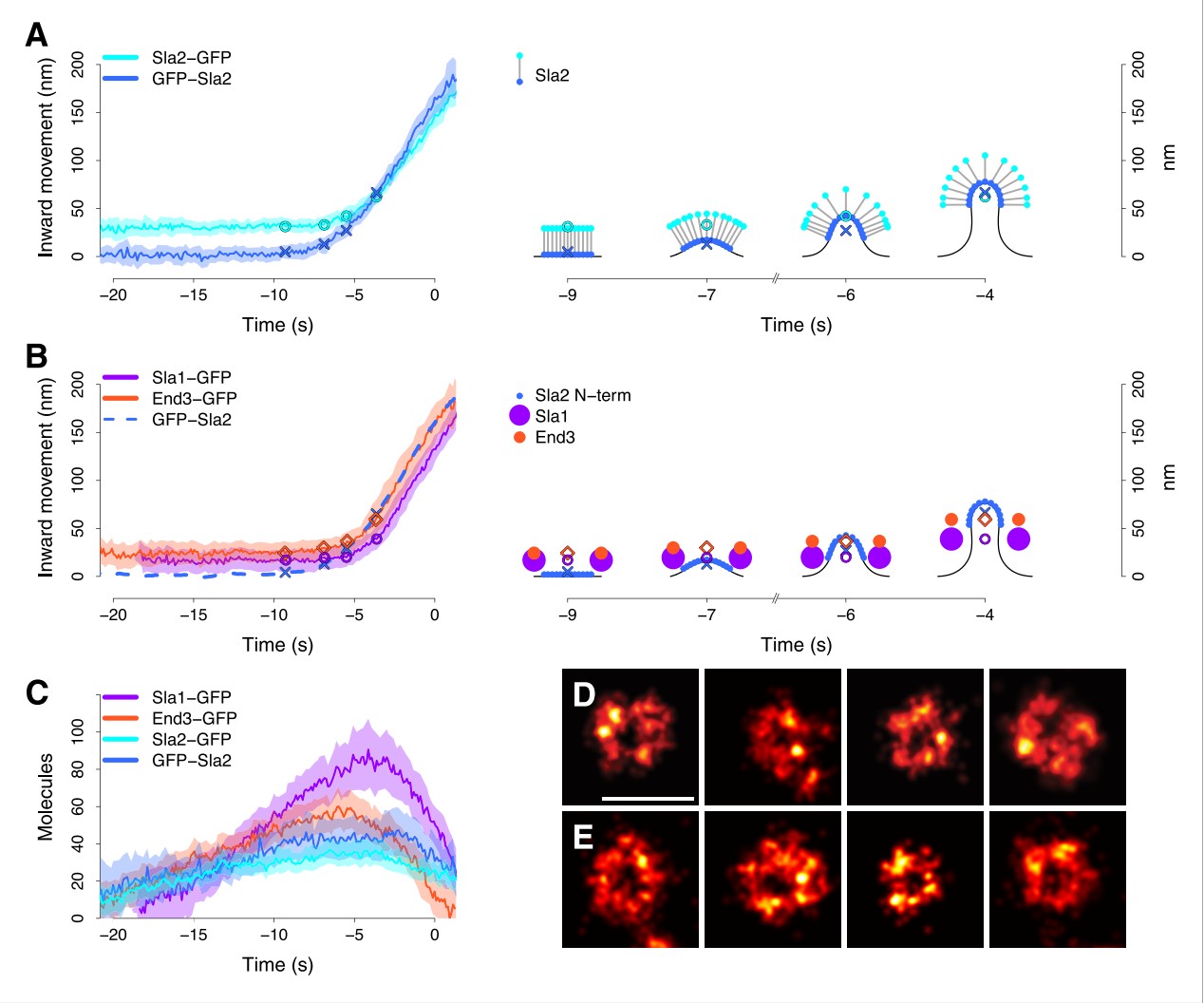

**Figure 3**. Coat dynamics and organization. (**A**) Left panel: the inward movement of Sla2 coat protein, tagged at its N- or C-terminus (GFP-Sla2 and Sla2-GFP respectively). Right panel: our model of Sla2 organization at the tip of the plasma membrane invagination. The centers of mass of the Sla2 model are marked by an 'X' (N-terminus) and an open circle (C-terminus). (**B**) Left panel: the inward movement of Sla1-GFP and End3-GFP. GFP-Sla2 (dashed line) is plotted for comparison. Right panel: our model of Sla1 and End3 organization at the outer rim of the coat. An open circle and an open diamond mark the centers of mass of the Sla1 and End3 models respectively. The center of mass of Sla2 N-terminus is marked by an 'X' and it is plotted for comparison. (**C**) The average number of molecules for GFP-Sla2, Sla2-GFP, Sla1-GFP and End3-GFP. (**D**) Sla1 ring structures imaged at endocytic sites, using superresolution microscopy. (**E**) Sla1 ring structures imaged at endocytic sites, using superresolution microscopy, in yeast cells treated with LatA. The shading of the trajectories represents the confidence interval. The plotted trajectories are listed in *Supplementary file 1*. Scale bar is 100 nm long. See also *Figure 3—figure supplement 1* and *Figure 3—figure supplement 2*.

The following figure supplements are available for figure 3:

**Figure supplement 1**. Organization of the coat and experimental control for the two color alignment.

**Figure supplement 2**. Imaging of Sla1 assemblies by localization microscopy.

a C-terminal SNAP tag and imaged structures on the bottom membrane of the cell close to the coverslip surface. With an average localization precision of approximately 10 nm, we could directly observe that Sla1 was localized in ring shapes (*Figure 3D* and *Figure 3—figure supplement 2A,B,E*). We also observed some less defined shapes, which likely correspond to states either during coat assembly or disassembly (*Figure 3—figure supplement 2C,D,F,G*). To exclude the possibility that a ring was observed because we were imaging a curved membrane, we treated Sla1-SNAP expressing

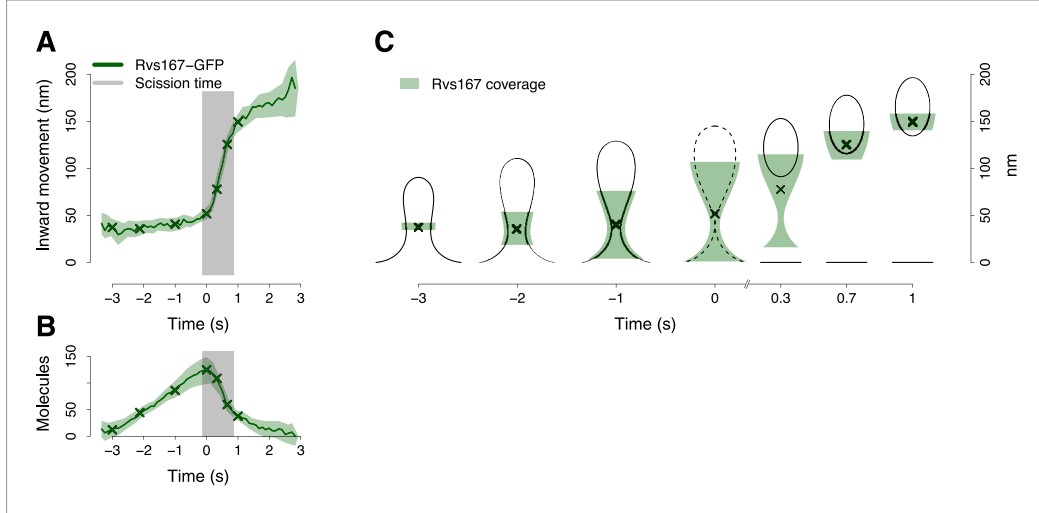

**Figure 4**. BAR protein dynamics. (**A**) The inward movement of Rvs167-GFP. (**B**) The average number of molecules of Rvs167-GFP. (**C**) Our model of Rvs coverage of the plasma membrane invagination during the invagination growth. When scission happens, Rvs molecules are released rapidly and remain in proximity of the vesicle only. The centers of mass of the Rvs model are marked by an 'X'. The grey vertical bar represents the estimated time window during which scission happens (*Kukulski et al., 2012*). The shading represents the confidence interval. The plotted trajectories are listed in *Supplementary file 1*. See also *Figure 4—figure supplement 1*.
The following figure supplement is available for figure 4:

**Figure supplement 1**. The variability in the BAR protein coverage of the plasma membrane.

cells with Latrunculin A, which inhibits actin polymerization and prevents membrane bending (*Kukulski et al., 2012*). Again, we observed ring-shaped Sla1 structures (*Figure 3E*) even though the plasma membrane remained flat (*Figure 3—figure supplement 2H*). These results show that Sla1 is organized in a circular pattern at the endocytic site *prior* to membrane invagination.

## Assembly of Rvs proteins at the neck of the invagination

The BAR domain proteins Rvs161 and Rvs167 form stable heterodimers, which localize transiently to the neck region of the endocytic invagination where they are thought to regulate vesicle scission (*Kaksonen et al., 2005*; *Ren et al., 2006*; *Idrissi et al., 2008*; *Youn et al., 2010*; *Kishimoto et al., 2011*; *Kukulski et al., 2012*). We thus aimed to better characterize the dynamics of the Rvs proteins during vesicle budding.

The average trajectory of Rvs167-GFP showed three different phases (*Figure 4A*). During the first phase the centroid moved inward linearly less than 15 nm while the Rvs167-GFP molecules assembled at a constant rate of ~40 molecules/s (*Figure 4A,B*). In the next phase the centroid moved rapidly for ~100 nm (*Figure 4A*) in less than a second. This rapid centroid movement coincided with a transition to disassembly and with the scission of the vesicle (*Figure 4B*). In the last phase the centroid continues to move inward, but with a reduced rate (*Figure 4A*). The fast centroid movement of the Rvs proteins has been described before, but its mechanism remained unknown (*Kaksonen et al., 2005*).

To understand better how the Rvs molecules are assembled on the membrane we supplemented our tracking data with data from structural studies of BAR domain proteins homologous to Rvs (*Peter et al., 2004*; *Mizuno et al., 2010*; *Mim et al., 2012*). These studies have shown that BAR domains can form regular oligomeric structures on membrane tubules of varying diameters, giving us an estimate of the density of dimers on the membrane. With our estimate of the numbers of Rvs167 molecules (*Figure 4B*), such density yields the possible membrane area covered by the Rvs dimers, assuming that all dimers are membrane bound. We then used the Rvs167 average trajectory to locate the area of the membrane profiles covered by Rvs. These simple assumptions allowed us to generate a dynamic model of the membrane area covered by the Rvs molecules (*Figure 4C*; see 'Materials and methods'; see also *Arasada and Pollard, 2011* for a similar approach). To obtain an estimate of Rvs

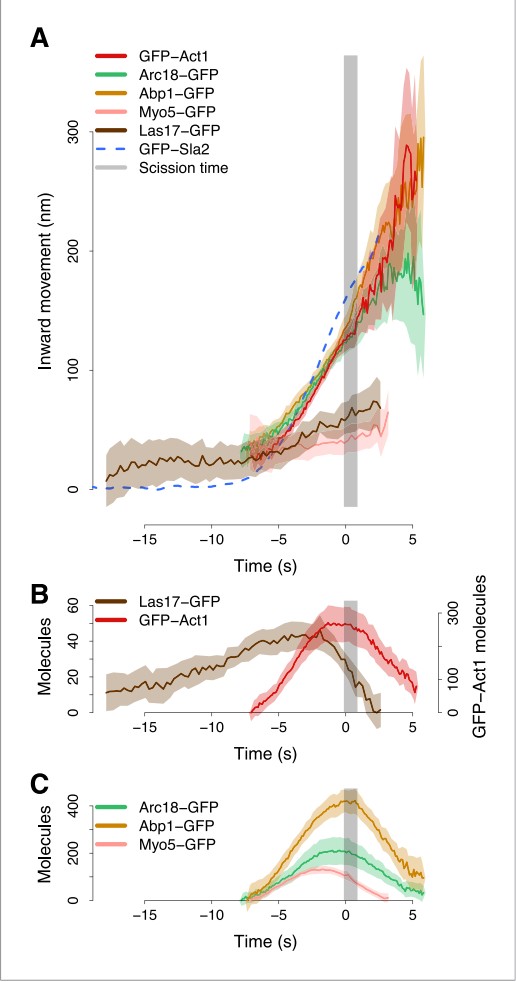

**Figure 5**. Actin cytoskeleton dynamics. (**A**) The inward movement of the actin cytoskeleton components GFP-Act1, Arc18-GFP and Abp1-GFP, together with the nucleation factors Las17-GFP and Myo5-GFP. GFP-Sla2 (dashed line) is plotted for comparison. (**B–C**) The average number of molecules of Las17-GFP, GFP-Act1, Myo5-GFP, Abp1-GFP and Arc18-GFP. The grey vertical bar represents the estimated time window during which scission happens (*Kukulski et al., 2012*). The shading represents the confidence interval. The plotted trajectories are listed in *Supplementary file 1*.

coverage immediately prior to scission we extrapolated the coordinates of the invagination profile at time 0 from the available membrane profiles.

This reconstruction of Rvs localization revealed the dynamics of Rvs assembly and disassembly in relation to membrane invagination and scission (*Figure 4C*). Rvs proteins start assembling at the midpoint of the endocytic invagination and the Rvs covered membrane area increases linearly and reaches its maximum at about the time of scission. Immediately prior to scission there are enough Rvs molecules to cover the whole tubular part of the invagination from the base of the invagination to the coated tip (*Figure 4C*; time point 0). After reaching their peak number, the Rvs molecules disassemble rapidly (*Figure 4B*) and their center of mass moves quickly inward toward the region where the newly formed vesicle is located. Our reconstruction suggests that the rapid movement of the centroid is the result of a rapid disassembly of the Rvs molecules at the neck of the invagination, where scission occurs (*Kukulski et al., 2012*), while some of the Rvs molecules are kept at the vesicle surface where they stay ~2 s longer.

## Assembly of the actin cytoskeleton

Polymerization of actin filaments at the endocytic site is critical for membrane invagination and vesicle budding (*Merrifield et al., 2002*; *Kaksonen et al., 2003*; *Boulant et al., 2011*; *Idrissi et al., 2012*; *Kukulski et al., 2012*). The endocytic actin filaments are nucleated by the Arp2/3 complex, which binds to existing filaments and nucleates new filaments in a branched configuration. Thus, actin filaments are organized as a dense, branched network, which surrounds the endocytic site and the newly formed vesicle (*Mulholland et al., 1994*; *Idrissi et al., 2012*; *Kukulski et al., 2012*).

To gain further insight into the assembly of the endocytic actin patches we tracked Act1, the yeast actin protein, and Arc18, a component of the Arp2/3 complex. Arc18 was GFP-tagged C-terminally at its genomic locus. As the GFP fusion of actin is not fully functional we expressed GFP-Act1 in addition to the endogenously expressed untagged Act1 (*Doyle and Botstein, 1996*). In actin patches the GFP-Act1 localization was comparable to general actin staining (*Kaksonen et al., 2003*) and the lifetime of Abp1-mCherry patches was unaffected (*Figure 1—figure supplement 2A*). GFP-Act1 is thus likely to be a good marker for actin localization and assembly at the endocytic site. We also tracked Las17, the yeast homolog of mammalian N-WASP, and Myo5, a type I myosin, which were both C-terminally GFP-tagged at their genomic loci. Las17 and Myo5 can activate the Arp2/3 complex and are critical for the initiation of actin polymerization and for membrane invagination (*Sun et al., 2006*).

The average trajectories of both Act1 and Arc18 followed closely the trajectory of Abp1 (*Figure 5A*). All three average trajectories started at 30–40 nm above the level of the plasma membrane when Sla2 movement began. Their centroids moved for ~100 nm until the scission time (*Figure 5A*). The tight colocalization of the centroids of Act1, Arc18 and Abp1 suggests that they are

homogeneously distributed within the actin network. After scission the centroid trajectories became noisier (*Figure 5A*). The increased noise probably reflects the free diffusion of the actin-covered vesicle in the cytoplasm (*Berro and Pollard, 2014*).

The trajectories of individual Las17-GFP and Myo5-GFP patches did not exhibit any clear inward movement. To align these trajectories with respect to the invagination axis we thus used two-color imaging of Las17-GFP or Myo5-GFP with Abp1-mCherry, and we aligned the Las17-GFP and Myo5-GFP trajectories based on the alignment of the corresponding Abp1-mCherry trajectories to Abp1-GFP average trajectory (see 'Materials and methods').

At the beginning of the membrane invagination Las17 and Myo5 average trajectories are close to Act1, Arc18 and Abp1 (*Figure 5A*). The Myo5 trajectory remained almost stationary during the invagination of the plasma membrane, whereas the Las17 trajectory moved inward, but much less than Abp1, Arc18 and Act1 trajectories.

The amount of Las17 molecules reached a maximum of ~45 molecules. At ~2 s before scission the amount of Las17 molecules started decreasing while the number of Act1 molecules reached a plateau which was maintained until the scission event (*Figure 5B*). Also the amount of Arc18 and Myo5 molecules peaked at ~2 s before scission. However, the amount of Myo5 molecules started decreasing before scission, while Arc18, similarly to Act1, maintained a plateau level until scission. The amount of Abp1 molecules peaked at the scission time and then rapidly declined (*Figure 5C*). We measured by quantitative Western blotting that GFP-Act1 represented $9 \pm 1\%$ of the total cellular actin (See 'Materials and methods'). If we assume that GFP-Act1 is recruited to endocytic sites with the same efficiency as the untagged actin, we can estimate that the peak number of actin molecules at the endocytic site is ~3000.

These results document the coupling between the assembly dynamics of the actin cytoskeleton and the changes in membrane shape. However, the homogeneous distribution of the actin cytoskeleton components across the actin network did not allow us to resolve its organization. To understand how actin polymerization is distributed over the network, we next combined centroid tracking with local photobleaching.

## Actin polymerizes at the base of the invagination

The mechanism by which the polymerizing actin filaments contribute to membrane invagination is poorly understood. Different models have been suggested, but there is no conclusive data in favor of any of them (*Kaksonen et al., 2003*; *Merrifield, 2004*; *Takenawa and Suetsugu, 2007*; *Suetsugu, 2009*; *Collins et al., 2011*; *Idrissi et al., 2012*; *Mooren et al., 2012*). *Figure 6A* summarizes three possible models for the organization of actin filaments at the endocytic site. In the first model the polymerizing ends of actin filaments are growing against the coat, thereby pushing it inward. In the second model filaments are growing against the plasma membrane at the base of the invagination, and the older filaments are connected to the coat to mediate the pushing force. In the third model actin polymerization is not preferentially oriented. In all models, the centroid of labeled actin would move similarly inward during endocytosis and thus could not be used to discriminate between the models. However, the models can be distinguished if the centroid is tracked after local photobleaching, because the location of the centroid is then determined by the fluorescence of the newly polymerized filaments. Model 1 predicts that after photobleaching the centroid would jump inward, away from the cell surface. Model 2 predicts a jump in the opposite direction, toward the cell surface, while model 3 predicts no jump. The centroid positions at later time points should be parallel to the reference trajectory, and be shifted by the distance of the post-bleach jump (*Figure 6A*).

We used a laser beam focused on a ~0.5 μm diameter spot to photobleach an individual fluorescence patch at the cell equator (*Figure 6B*) and we tracked its centroid position before and after photobleaching. Multiple trajectories were averaged and plotted with the average trajectory of the unbleached patch, as a comparison. We first photobleached GFP-Act1 patches ~3–4 s after their appearance. After photobleaching the reappearing centroids were located closer to the plasma membrane than the centroids at the corresponding time point in the unbleached average trajectory (*Figure 6C*). Neighbor trajectories to the photobleaching site, which were only partially affected by photobleaching, did not show any jump (*Figure 6—figure supplement 1A*). We next photobleached Arc18-GFP and observed a similar, although slightly smaller, shift of the centroid position toward the plasma membrane (*Figure 6D*). These results suggest that actin filament nucleation and polymerization take place at the base of the invagination, in agreement with model 2. In addition, we

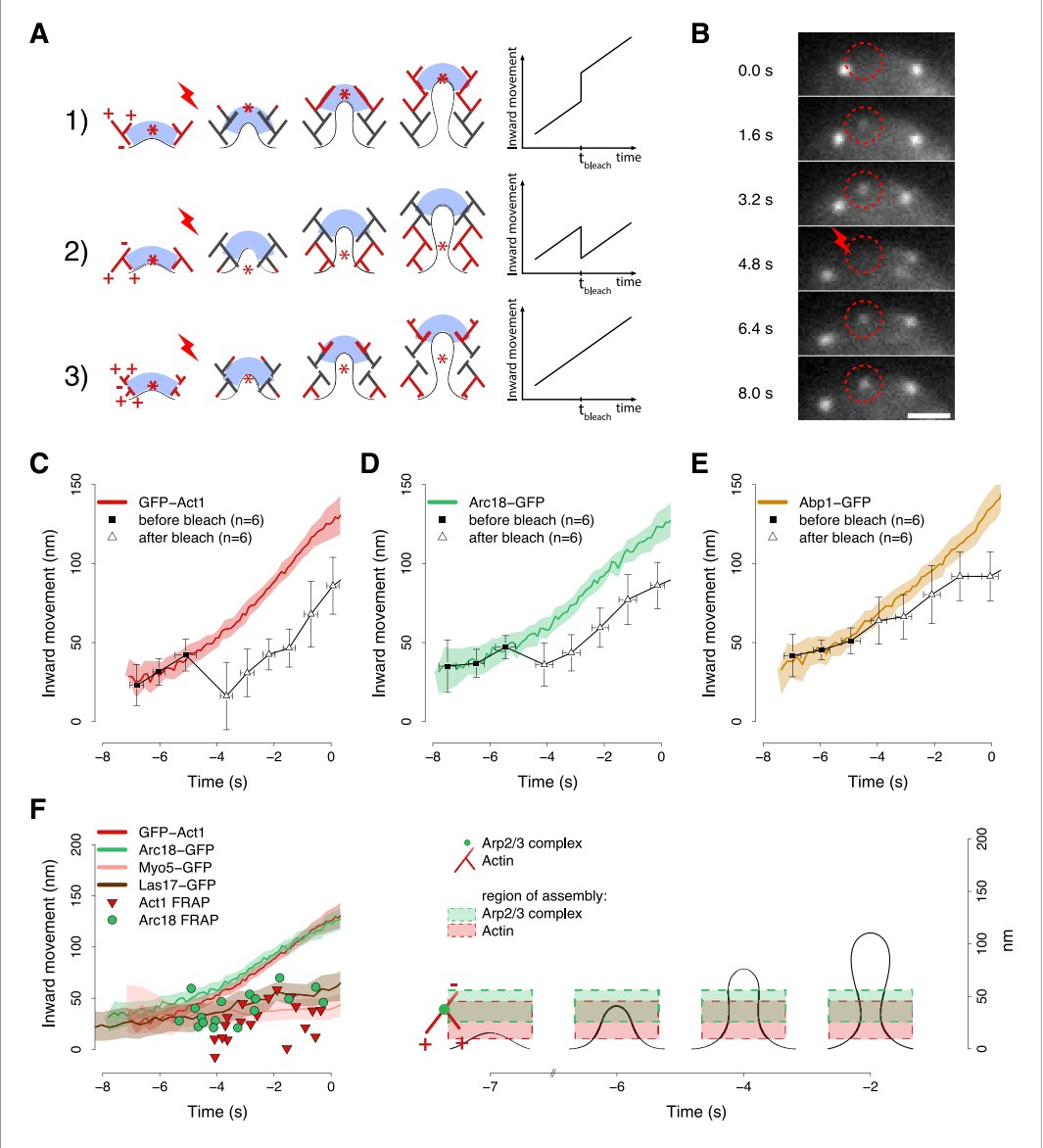

**Figure 6**. Region of assembly of actin filaments. (**A**) A schematic cartoon to show how local photobleaching would affect the centroid position of a fluorescent actin patch, given three possible scenarios for the nucleation of new actin filaments (in red) at the endocytic locus: (1) New actin filaments are nucleated in the proximity of the coat; photobleaching would shift the centroid away from the plasma membrane. (2) New filaments are nucleated at the base of the plasma membrane invagination; photobleaching would shift the centroid toward the plasma membrane. (3) There is no preferential direction for actin nucleation; photobleaching would not affect the centroid position. (**B**) Local photobleaching of an endocytic event in a cell expressing GFP-Act1. Scale bar: 1 μm. See also *Video 1*. (**C–E**) Centroid positions of endocytic patches in yeast cells expressing GFP-Act1 (**C**), Arc18-GFP (**D**) or Abp1-GFP (**E**) before and after photobleaching. Error bars represent the Standard Deviation. Plots show the respective average trajectories for comparison. (**F**) Left panel: The locations of the first post-bleach centroids of Act1-GFP (red triangles) and Arc18-GFP (green circles) aligned relative to the unbleached average trajectories. Nucleation promoting factors, Myo5-GFP and Las17-GFP, localize to the same region where actin monomers are added. Average trajectories of GFP-Act1 and Arc18-GFP are plotted for comparison. Right panel: Our model for the region of actin polymerization. The dashed boxes highlight the region where actin and Arp2/3 complexes are recruited. The centers and the heights of the boxes correspond respectively to the average and to the standard deviation of the centroid positions of GFP-Act1 and Arc18-GFP after photobleaching. The cartoon highlights the regions where the new Arp2/3 complexes and actin molecules

*Figure 6. continued on next page*

*Figure 6. Continued*

are recruited. The shading of the trajectories represents the confidence interval. The plotted trajectories are listed in *Supplementary file 1*. See also *Figure 6—figure supplement 1*.

The following figure supplement is available for figure 6:

**Figure supplement 1**. The position of Abp1 fluorescence recovery after photobleaching.

---

photobleached Abp1-GFP, a proposed inhibitor of actin filament nucleation (*D'Agostino and Goode, 2005*). However, the trajectories of the Abp1-GFP centroid with or without photobleaching did not differ significantly (*Figure 6E*) suggesting that Abp1 molecules are recruited throughout the actin network unlike actin and the Arp2/3 complex.

We next performed similar photobleaching experiments at different time points during endocytosis, and recorded the centroid position right after local photobleaching (*Figure 6F*). At all time, GFP-Act1 and Arc18-GFP were added at the base of the invagination, but the exact localizations of the two proteins differed: Act1 was added on average at 27 ± 18 nm (mean ± SD, n = 19) and Arc18 at 41 ± 15 nm (mean ± SD, n = 17) above the plasma membrane surface (p value = 0.02). We note that this is also where the Arp2/3 activators Las17 and Myo5 are located, according to the average centroid trajectories (*Figure 6F*). The post bleach centroid position of Abp1 followed closely the Abp1 reference trajectory (*Figure 6—figure supplement 1B*).

These results suggest that both actin filament nucleation and polymerization take place at the base of the endocytic invagination. The Arp2/3 complex is assembled slightly higher above the membrane plane compared to actin, which is consistent with the dendritic nucleation model (*Pollard and Borisy, 2003*) where the actin plus ends are oriented toward the plasma membrane (*Figure 6F*). The putative inhibitor of actin nucleation, Abp1, would be distributed uniformly within the network.

## Discussion

We developed an approach based on centroid tracking and averaging to characterize the movement and assembly dynamics of proteins in the endocytic machinery during vesicle budding. We applied this approach to endocytic proteins representing different functional modules to generate a comprehensive model of the dynamic behavior of the endocytic machinery (*Figure 7A,B*). We combined the centroid tracking data with time-resolved protein abundances, with EM data about membrane shapes and with superresolution imaging of the protein distribution within the diffraction limited volume. This hybrid approach allowed us to visualize the endocytic machinery with both high temporal and spatial resolution. Conceptually similar approach for temporal alignment of endocytic patch dynamics has recently been independently described by *Berro and Pollard (2014)*.

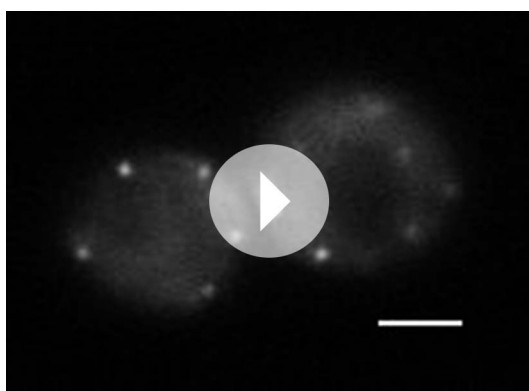

**Video 1.** Photobleaching experiment. Local photo-bleaching of an endocytic event in a cell expressing GFP-Act1. The video plays in real time. Scale bar is 2 μm.

### The endocytic coat

The organization of clathrin molecules in the endocytic coat is extensively documented (*Brodsky et al., 2001*; *Fotin et al., 2004*; *Kirchhausen, 2009*), but the organization of the numerous coat-associated adaptor and scaffold proteins is less well understood. However, several studies indicate that many of these proteins have distinct distributions within the coat (*Tebar et al., 1996*; *Saffarian and Kirchhausen, 2008*; *Boettner et al., 2011*). We showed that three coat-associated proteins exhibit specific orientation and distribution within the endocytic coat (*Figure 3*).

Sla2 molecules are oriented so that the N-terminal membrane binding-domain is at the plasma membrane, while the C-terminal actin-binding domain is ~30 nm away from the plasma

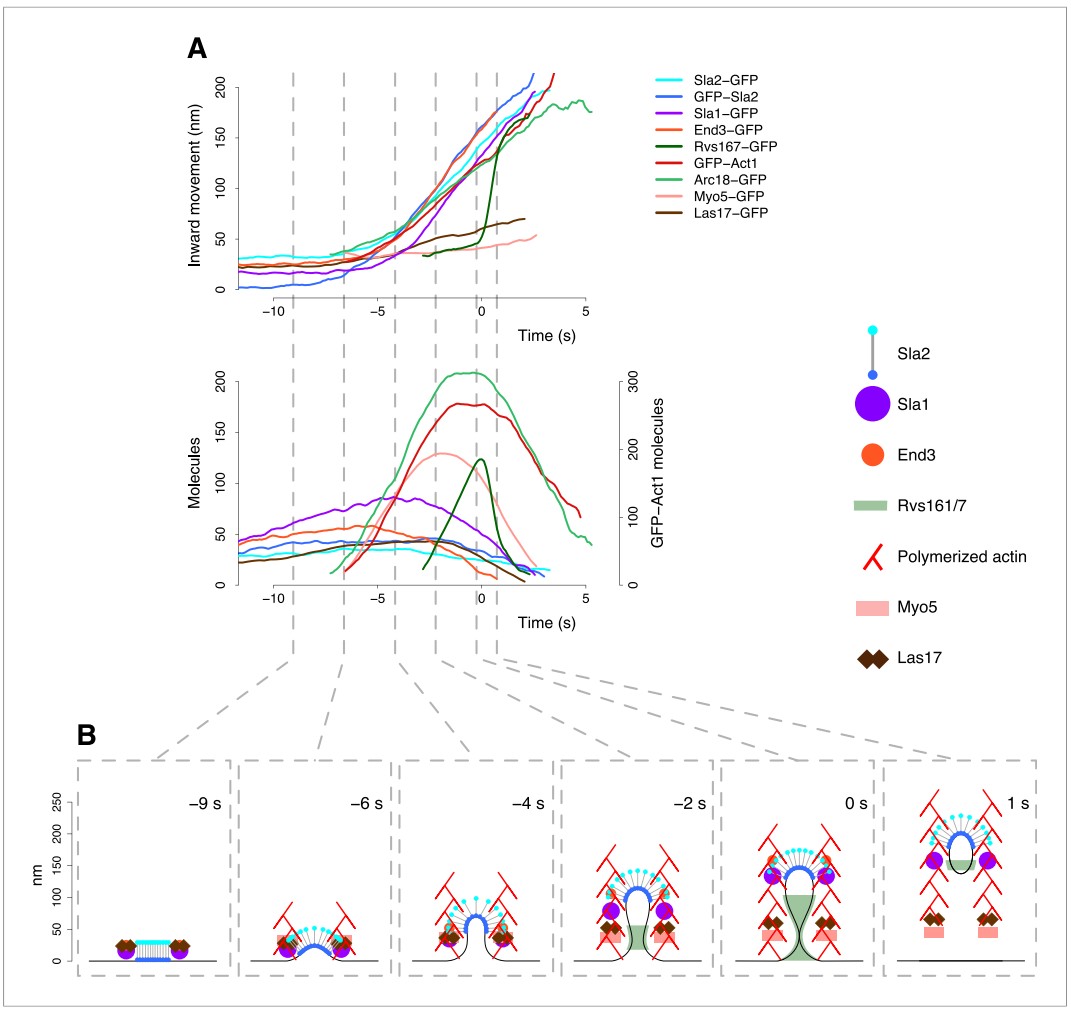

**Figure 7**. The dynamic architecture of the endocytic machinery. (**A**) Summary of the average dynamics and number of molecules for Sla2-GFP, GFP-Sla2, Sla1-GFP, End3-GFP, Rvs167-GFP, Arc18-GFP, Myo5-GFP, Las17-GFP and GFP-Act1. The plotted trajectories are listed in **Supplementary file 1**. Las17 and Myo5 were smoothened using a moving average filter of length 5; the remaining data were smoothed using a Savitzky-Golay filter over 11 time points. (**B**) The protein symbols are positioned, according to the average trajectories of the corresponding proteins (**Figure 7A**), along the plasma membrane invagination profiles derived from CLEM. The actin filament network is drawn schematically to illustrate the average direction of actin polymerization and does not represent the true organization of the actin filaments and branches. *Time ≤ −8 s*: coat assembly; Sla2 exposes its actin binding domain outside of the clathrin cage. *Time ≈ −8 s*: actin polymerization starts and drives the initiation of the plasma membrane invagination. *−8 s ≤ time ≤ −3 s*: the invagination grows and the coat accommodates the changes in curvature imposed by the plasma membrane invagination growth. *−3 s ≤ time < 0 s*: Rvs proteins are recruited to the invagination, which is growing longer, and stabilize it. *Time ≥ 0*: the scission of the invagination releases the vesicle. Concomitantly, the Rvs structure starts to disassemble very rapidly. See also **Video 2**.

membrane. The coiled-coil region separating the terminal domains contains a clathrin-binding motif (**Engqvist-Goldstein et al., 2001**; **Boettner et al., 2011**). Therefore, Sla2 molecules could span the clathrin lattice, which is about 22 nm from the membrane surface (**Vigers et al., 1986**), and interact simultaneously with the membrane, with clathrin, and with actin filaments on the cytoplasmic side of the clathrin lattice.

Sla1 and End3, together with a third protein Pan1, form a complex, which is implicated in the regulation of actin assembly (**Holtzman et al., 1993**; **Bénédetti et al., 1994**; **Tang et al., 2000**). Sla1 is localized in a ring of ~50 nm in diameter, which can fit well around the tip of the endocytic membrane invagination of ~30 nm in diameter (**Idrissi et al., 2008**; **Kukulski et al., 2012**). As Sla1 and End3 are late-assembling coat proteins (**Kaksonen et al., 2005**; **Newpher et al., 2005**), they could assemble at

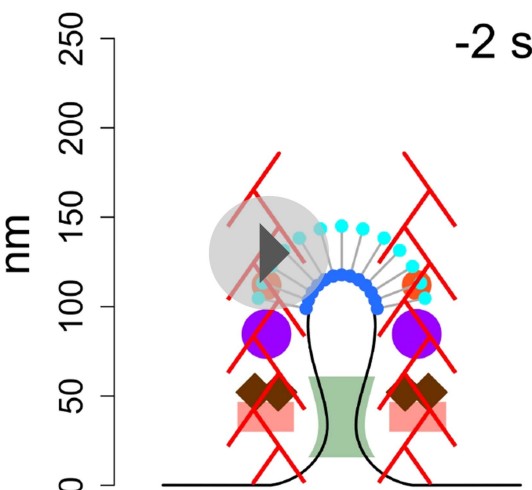

**Video 2.** The dynamic architecture of the endocytic machinery. The reconstruction of the dynamic architecture of the endocytic machinery obtained by combining the centroid positions of the proteins over time, the plasma membrane invagination profiles derived from CLEM (*Kukulski et al., 2012*) and the structural properties of the proteins. See also *Figure 7* for the legend of the protein symbols.

the rim of a pre-existing early coat. The disassembly of Sla1 and End3 begins after membrane invagination has started, but several seconds before scission, suggesting that these proteins may be important during early stages of vesicle budding. Interestingly, Sla2 and Pan1, have been suggested to capture short cytosolic actin filaments that could serve as 'mother filaments' to recruit the Arp2/3 complex and to initiate the assembly of the actin network (*Chen and Pollard, 2013*).

Taken together, our data demonstrate that the endocytic coat exhibits a highly organized and dynamic molecular architecture during vesicle budding.

## Rvs BAR-domain proteins

BAR domain proteins can sense membrane curvature and they can actively induce membrane curvature at high concentration (*Peter et al., 2004*; *Mizuno et al., 2010*; *Mim et al., 2012*). In yeast, the heterodimeric BAR domain proteins Rvs161/167 assemble transiently at the endocytic site during the formation of the membrane invagination (*Kaksonen et al., 2005*; *Idrissi et al., 2008*; *Kukulski et al., 2012*) to regulate the invagination or scission steps (*Kaksonen et al., 2005*; *Youn et al., 2010*; *Kishimoto et al., 2011*; *Boucrot et al., 2012*; *Kukulski et al., 2012*). However, the exact mechanism of action of Rvs is unknown. Our results suggest that there are enough Rvs molecules to form a dense BAR domain lattice covering the whole tubular part of the invagination below the coated tip (*Figure 4*). At this density the Rvs proteins could exert significant bending forces on the membrane (*Sorre et al., 2012*). This Rvs lattice could therefore support the membrane tubule and prevent premature fission (*Kukulski et al., 2012*). In addition, an Rvs lattice could act as a lipid diffusion barrier that could form a lipid phase boundary to promote vesicle scission (*Liu et al., 2006*; *Zhao et al., 2013*). Furthermore, our model explains the rapid movement of the Rvs167-GFP centroid upon scission: due to their affinity for curved membranes, the Rvs molecules are likely to first disassemble at the tubular part of the invagination that collapses after scission, while some of the Rvs molecules may remain bound to the membrane that forms the vesicle. Therefore, the center of mass of Rvs molecules would rapidly shift due to localized disassembly without actual movement of the Rvs lattice. The exact timing of the Rvs disassembly, in respect to the scission event, remains an interesting open question. Currently, our data does not have the time resolution to discern whether the initiation of Rvs disassembly precedes scission or follows it. If Rvs disassembly follows scission it could be a passive response to the collapse of the membrane tubule. Alternatively, if Rvs disassembly precedes scission it might destabilize the membrane tubule and thereby regulate the timing of scission. In either case, the extensive Rvs lattice could support the membrane tubule and thereby promote its extension until the scission.

## Endocytic actin network

Actin nucleation and polymerization are precisely coordinated with membrane shape changes. Our previous CLEM results suggested that the initiation of membrane bending coincides with actin polymerization (*Kukulski et al., 2012*). Another EM study, however, suggested that the membrane bending begins already ~15 s prior to actin polymerization (*Idrissi et al., 2012*). The average trajectory of the coat proteins Sla2 shows directed inward movement only after actin polymerization has started (*Figure 1*) supporting the idea that the main membrane bending phase starts only after actin polymerization. However, a very shallow initial membrane curvature might be unresolvable by the tracking approach.

The data about numbers of molecules allow us to estimate some key parameters of the endocytic actin network. If each Arp2/3 complex nucleates an actin filament, new filaments are generated at a rate of ~30 filaments per second, reaching a maximum number of ~200 filaments. With the estimated maximum of ~3000 actin molecules the average filament length would be ~40 nm. The actin polymer, the Arp2/3 complex, and the Arp2/3 activators, Myo5 and Las17, reach the maximum level of molecules about 2 s before vesicle scission. Actin and the Arp2/3 complex maintain their levels of molecules until scission, whereas Las17 and Myo5 start to disassemble right after reaching their peak levels. These data suggest that the rate of nucleation of new actin filaments decreases before vesicle scission occurs. The initiation of actin filament nucleation has been studied extensively (*Mooren et al., 2012*), but the mechanisms that terminate nucleation are not well understood. The termination of nucleation could involve feedback from membrane shape or tension.

Different models have been proposed for the organization of actin filaments at the endocytic site (*Takenawa and Suetsugu, 2007*; *Mooren et al., 2012*). However, conclusive data about the location of nucleation and polymerization during endocytosis has not been available. We showed here that actin filaments are nucleated and polymerize at the base of the invagination, where the Arp2/3 activators Las17 and Myo5 are also localized (*Figure 6*). Capping proteins, which bind to the growing ends of actin filaments and stop polymerization (*Mooren et al., 2012*), are likely important for restricting actin polymerization to the base of the invagination.

## The functional organization of the endocytic machinery

By combining all our data we built a comprehensive model that reveals the dynamic interplay of the functional modules of the endocytic machinery during the final ~10 s of endocytosis (*Figure 7*). During this time window the flat plasma membrane is reshaped into an endocytic vesicle. The coat, which is initially assembled on flat membrane, starts internalizing and reorganizing when actin polymerization begins. Actin polymerization is initiated close to the plasma membrane where Las17 is located and the region of polymerization remains at the base of the invagination throughout vesicle budding. When the invagination has formed, actin is likely polymerizing in a circular region surrounding the invagination. Initially, actin polymerization might also be triggered in a circular region at the rim of the coat, where Sla1, a regulator of actin nucleation, is also located. Continued filament nucleation and polymerization at the plasma membrane would push the older filaments into the cell generating a flow of actin filaments. The organization of the endocytic actin network bears striking similarity to the organization of actin at the leading edge of migrating animal cells where actin filaments nucleated by the Arp2/3 complex polymerize and push against the plasma membrane (*Pollard and Borisy, 2003*). This polymerization pushes the leading edge of the cell forward, but also results in a retrograde flow of actin filaments into the cell. The actin binding domains of Sla2 are ideally positioned at the surface of the endocytic coat to capture the actin filaments that are moving away from the plasma membrane. We suggest that the actin scaffold is rigid enough to transmit the forces produced by the polymerization at the plasma membrane so that Sla2 can, together with Ent1 (*Skruzny et al., 2012*), transmit the force from the actin polymerization to the membrane.

Detailed quantitative analyses of assembly and disassembly of the endocytic machinery, especially the actin cytoskeletal proteins, have been previously performed on fission yeast *Schizosaccharomyces pombe* (*Wu and Pollard, 2005*; *Berro et al., 2010*; *Sirotkin et al., 2010*; *Arasada and Pollard, 2011*; *Chen and Pollard, 2013*). Comparing the *S. pombe* and *S. cerevisiae* data shows that the overall assembly dynamics and the stoichiometries of the endocytic proteins are highly similar in these very distantly related fungal species. For example, the numbers of the *S. pombe* homologs of Las17 and Myo3/5 also peak briefly before the peak of actin molecules (*Berro et al., 2010*; *Sirotkin et al., 2010*), suggesting that the assembly dynamics of the actin cytoskeleton are functionally optimized. However, there are also some clear differences: in *S. pombe* the Las17 homolog Wsp1 is recruited together with myosin just prior to actin polymerization (*Berro et al., 2010*; *Sirotkin et al., 2010*; *Arasada and Pollard, 2011*) whereas in *S. cerevisiae* Las17 localizes >10 s before myosins are recruited and actin polymerization starts (*Kaksonen et al., 2003*; *Figure 5*). Analyzing these evolutionary differences in detail could reveal mechanistic insights into the regulation of the endocytic assembly.

Rvs assembly and disassembly are tightly coupled to the membrane shape changes (*Figure 4*). Rvs molecules may get recruited by a specific membrane curvature at the invagination, but protein–protein interactions may also have a role in the recruitment (*Ren et al., 2006*). Just before scission, the membrane invagination is densely covered by Rvs molecules. Scission occurs within the

**Table 2**. Yeast strains

| Strain # | Genotype |
| --- | --- |
| MKY0216 | MATa, his3-Δ200, leu2-3,112, ura3-52, lys2-801, NUF2-EGFP::HIS3MX6 |
| MKY0217 | MATα, his3-Δ200, leu2-3,112, ura3-52, lys2-801, NUF2-EGFP::HIS3MX6 |
| MKY0711 | MATa, his3-Δ200, leu2-3,112, ura3-52, lys2-801, MYO5-EGFP::HIS3MX6, ABP1-mCherry::kanMX4 |
| MKY0822 | MATa, his3-Δ200, leu2-3,112, ura3-52, lys2-801, SLA1-EGFP::HIS3MX6, ABP1-mCherry::kanMX |
| MKY1304 | MATa, his3-Δ200, leu2-3,112, ura3-52, lys2-801, END3-EGFP::HIS3MX6, ABP1-mCherry::kanMX4 |
| MKY1318 | MATa, his3-Δ200, leu2-3,112, ura3-52, lys2-801, RVS167-EGFP::HIS3MX6, ABP1-mCherry::kanMX4 |
| MKY1368 | MATα, his3-Δ200, leu2-3,112, ura3-52, lys2-801, LAS17-EGFP::HIS3MX6, ABP1-mCherry::kanMX4 |
| MKY2119 | MATα, his3-Δ200, leu2-3,112Δ::GalL-ISce1-natNT2, ura3-52, lys2-801, sfGFP-SLA2-mCherry::hphNT1 |
| MKY2653 | MATa, his3-Δ200, leu2-3,112, ura3-52, lys2-801 with pMK0100[CEN, URA3 GFP-ACT1] |
| MKY2655 | MATa, his3-Δ200, leu2-3,112, ura3-52, lys2-801, ABP1-mCherry::kanMX4 with pMK0100[CEN, URA3 GFP-ACT1] |
| MKY2689 | MATα, his3-Δ200, leu2-3,112Δ::GalL-ISce1-natNT2, ura3-52, lys2-801, sfGFP-SLA2 |
| MKY2720 | MATa, his3-Δ200, leu2-3,112, ura3-52, lys2-801, ARC18-myEGFP::natNT2, ABP1-mCherry::kanMX |
| MKY2747 | MATa, his3-Δ200, leu2-3,112, ura3-52, lys2-801, ARC18-myEGFP::natNT2 |
| MKY2832 | MATa, his3-Δ200, leu2-3,112, ura3-52, lys2-801, RVS167-EGFP::HIS3MX6 |
| MKY2833 | MATa, his3-Δ200, leu2-3,112, ura3-52, lys2-801, SLA1-EGFP::HIS3MX6 |
| MKY2834 | MATα, his3-Δ200, leu2-3,112, ura3-52, lys2-801, ABP1-EGFP::HIS3MX6 |
| MKY2836 | MATα, his3-Δ200, leu2-3,112Δ::GalL-ISce1-natNT2, ura3-52, lys2-801, sfGFP-SLA2, ABP1-mCherry::kanMX |
| MKY3135 | MATα, his3-Δ200, leu2-3,112Δ::GalL-ISce1-natNT2, ura3-52, lys2-801, sfGFP-SLA2, ABP1-mCherry::kanMX, Sla1-SNAP::HIS3MX6 |
| MKY2859 | MATa, his3-Δ200, leu2-3,112, ura3-52, lys2-801, SLA2-EGFP::HIS3MX6 |
| MKY2863 | MATa, his3-Δ200, leu2-3,112, ura3-52, lys2-801, CSE4-EGFP::HIS3MX6 |
| MKY2864 | MATa, his3-Δ200, leu2-3,112, ura3-52, lys2-801, LAS17-EGFP::HIS3MX6 |
| MKY2876 | MATa, his3-Δ200, leu2-3,112, ura3-52, lys2-801, MYO5-EGFP::HIS3MX6 |

*Table 2. Continued on next page*

Rvs covered region at ~1/3 of the invagination length (Kukulski et al., 2012). However, the exact molecular mechanism of vesicle scission is still not understood. Neither Rvs, nor the yeast homolog of dynamin Vps1, are essential for scission, but they may regulate its timing or location (Kishimoto et al., 2011; Kukulski et al., 2012; Smaczynska-de Rooij et al., 2012). Interestingly, the location of actin polymerization and Myo5 corresponds closely to the site of scission, therefore polymerization and motor activities could have a role in scission (Jonsdottir and Li, 2004). After scission the Rvs molecules that remain on the newly formed vesicle might have additional roles, for example, in uncoating as shown for the mammalian homolog endophilin (Milosevic et al., 2011).

## Materials and methods

### Strains and plasmids

Yeast strains (Table 2) were generated by homologous recombination of the target genes with PCR cassettes. C-terminal tagging was performed using plasmids pFA6a-EGFP-His3MX6, pFA6a-mCherry-KanMX4 and pYM12-PKS134 for monomeric GFP (myEGFP). pMaM17 (Khmelinskii et al., 2012) was used for the tandem tag of mCherry and sfGFP. pMK03-SNAP-His3MX6 was generated by replacing EGFP with SNAPf (Sun et al., 2011) in pFA6a-EGFP-His3MX6. N-terminal tagging was performed using plasmid pMaM173 which was transformed into yeast strains with a Gal L - ISCE1 integration used to loop out the URA marker and the TEF promoter (Khmelinskii et al., 2011). pMaM175 was used to tag Nuf2 C-terminally with sfGFP (Khmelinskii et al., 2011). The strains used for protein abundance measurements were confirmed by sequencing of the integrated tags.

### Live-cell imaging

Yeast cells were grown to logarithmic phase on SC-Trp medium at 25°C. They were adhered to ConA coated coverslips. Cells were incubated for 10 min at room temperature on the ConA coated coverslip and then washed with SC-Trp medium. Cells were imaged on the coverslip in 40 μl of SC-Trp medium. All samples were imaged at room temperature using an Olympus IX81 wide-field epifluorescence microscope equipped with a 100×/1.45 objective. For single channel live cell imaging 488 nm laser light and images were acquired with 80–100 ms exposure time. Emission light was filtered using the GFP-3035C-OMF single-band filter set

*Table 2. Continued*

| Strain # | Genotype |
|---|---|
| MKY2880 | MATa, his3-Δ200, leu2-3,112, ura3-52, lys2-801, ABP1-mCherry-sfGFP::kanMX4 |
| MKY2893 | MATa, his3-Δ200, leu2-3,112, ura3-52, lys2-801, END3-EGFP::HIS3MX6 |
| MKY2918 | MATα, his3-Δ200, leu2-3,112, ura3-52, lys2-801, SLA2-EGFP::HIS3MX6, ABP1-mCherry::kanMX4 |
| MKY2919 | MATa, his3-Δ200, leu2-3,112Δ::GalL-ISce1-natNT2, ura3-52, lys2-801, NUF2-sfGFP::KIURA3 |
| MKY2920 | MATα, his3-Δ200, leu2-3,112Δ::GalL-ISce1-natNT2, ura3-52, lys2-801, NUF2-sfGFP::KIURA3 |
| MKY3136 | MATa, his3-Δ1, leu2-Δ0, ura3-Δ0, met15-Δ0, LAS17-myEGFP::natNT2 |
| MKY3137 | MATa, his3-Δ1, leu2-Δ0, ura3-Δ0, met15-Δ0, RVS167-myEGFP::natNT2 |

The yeast strains used in this study.

(Semrock, Rochester, NY). Fluorescence was detected using the Hamamatsu ImagEM EMCCD camera. For two color live cell imaging the samples were excited simultaneously with 488 nm and 561 nm laser light for 250 ms exposure. Excitation light was reflected with a OBS-U-M2TIR 488/561 (Semrock, Rochester, NY) dichroic mirror. Emission light was split and filtered with the DUAL-view (Optical Insights, LLC, Tucson, AZ) beam splitter. The beam splitter created two separated images, one for each channel, on the Hamamatsu ImagEM EMCCD camera sensor. Photobleaching was performed using a custom built setup with a 488 nm laser, focused on a ∼0.5 µm spot.

The wide-field epifluorescence microscope setup was controlled by Metamorph 7.5 (Molecular Devices, Sunnyvale, CA).

## Image analysis

Before tracking the endocytic patches and the photobleaching experiments, the extracellular background was subtracted from the images using the Background Subtraction function in ImageJ (with rolling ball radius equal to 90 pixels, corresponding to 9 µm). To correct for photobleaching, each image was then scaled such that the average fluorescence intensity within the cell remains constant in successive frames. To correct for the uneven cytoplasmic background signal at the edge of the cell, we estimated the cytoplasmic contribution by further processing the images with a median filter (of kernel 6 pixels, corresponding to 0.6 µm). We then subtracted the cytoplasmic background and tracked endocytic patches using the Particle Tracker plugin in (*Sbalzarini and Koumoutsakos, 2005*).

## Single channel trajectory alignment and averaging

Each protein trajectory $p$ is a list of points,

$$p_i = \left\{ p_i^x, p_i^y, p_i^f \right\},$$

defined from $p_i^x$ and $p_i^y$, the centroid 2D coordinates in the focal plane, and $p_i^f$, the corresponding fluorescence intensity, where the index $i$ denotes time. Trajectories were aligned in space and time by a custom made software written in R (www.CRAN.org). An isometric transformation of space $T = \{T^x, T^y, T^\theta\}$ is defined as

$$T : \{x, y\} \rightarrow \left\{ \cos\left(T^\theta\right) x - \sin\left(T^\theta\right) y + T^x, \ \sin\left(T^\theta\right) x + \cos\left(T^\theta\right) y + T^y \right\}.$$

The best alignment between two trajectories, $p$ and $q$, is computed by minimizing the sum of the squared difference between all overlapping points:

$$\{T_{best}, \tau_{best}\} = \arg \min_{T, \tau} \left( \frac{\sum_i w_i \left( \left(q_{i+\tau}^x - (Tp_i)^x\right)^2 + \left(q_{i+\tau}^y - (Tp_i)^y\right)^2 \right)}{\sum_i w_i} \right), \ \text{with } w_i = q_{i+\tau}^f p_i^f.$$

In practice, we computed for each possible time shift $\tau$, the best spatial transformation (*Horn, 1987*), and selected the best overall result. The weights $w_i$ are the product of the fluorescence intensities of the spot pairs at each time point. Being proportional to the cross-correlation in time of the fluorescence intensities, they helped to refine the temporal alignment of the trajectories. To not bias the alignment by the choice of a reference, each trajectory was separately used as a reference to which all the remaining trajectories were aligned. Given $n$ trajectories we thus computed $n(n-1)$ alignments. Each alignment gave us an estimate of the transformation that aligns a trajectory to its reference. We thus estimated the average transformation that aligns all the trajectories together as

seen from a common reference point in the field of view. Once aligned, all trajectories were averaged to obtain the average trajectory. The average trajectory of each protein was derived from 50 to 80 individual trajectories (*Figure 1—figure supplement 1*, *Table 1*). See *Source code 1.*

The alignment of the trajectories of Abp1, Arc18 and Act1 was computed using only the trajectory data associated with the invagination dynamics, up to the trajectory peak in fluorescence intensity.

The number of molecules over time was obtained after calibrating the fluorescence intensity curve mean integral with the average number of molecules estimated at the endocytic spot (see 'Materials and Methods': calibration of the fluorescence intensity curve of the trajectories with the number of molecules).

All data presented in this work are listed in the *Supplementary file 1*. All the curves in *Figure 7*, except Las17 and Myo5, were smoothened using a Savitzky-Golay filter over 11 time points. Las17 and Myo5 were smoothened using a moving average filter of length 5.

## Two color alignment procedure

The alignment is performed using pairs of trajectories acquired simultaneously with two labeled proteins: Abp1-mCherry serving as a reference, and a protein of interest tagged with GFP. Simultaneous acquisition of the two colors is made with the DUAL-view beam splitter (Optical Insights, LLC, Tucson, AZ). Image un-splitting and correction of chromatic aberration are done following the DUAL-view recommended procedures. A sample with TetraSpeck microsphere (0.1 µm, Invitrogen, Carlsbad, CA) was imaged in both channels and the centroids were identified with Particle Tracker (*Sbalzarini and Koumoutsakos, 2005*) in ImageJ and then processed in MATLAB (TheMathworks, Natick, MA) to generate a spatial warping transformation that was applied on the raw coordinates. The non-linear transformation was generated by the local weighted mean (lwm) method of cp2tform (See Matlab help and references thereafter).

## Definitions

The average trajectory of a protein, $P$, is represented by a temporal collection of 6-dimensional vectors,

$$P_i = \left\{ P_i^x, P_i^y, P_i^f, \delta_i^x, \delta_i^y, \delta_i^f \right\} \text{ with } i = \{1, \ldots, N\}.$$

$P$ contains $P^x$ and $P^y$, the average position of the centroid of the fluorescence patch, and the corresponding average fluorescence $P^f$, the standard error of the means $\delta^x$ and $\delta^y$, associated with the position coordinates, and $\delta^f$, the standard error of the mean associated with the fluorescence intensity. The average trajectory of the reference protein, that we name $R$, follows the same notation.

The trajectory pairs are not necessarily defined over the same time interval as the average trajectories $P$ and $R$. They were thus smoothed with a moving average, to reduce noise, and interpolated with a cubic spline to estimate values at the missing time points. The resulting trajectory, $p$ is a temporal collection of 3-dimensional vectors

$$p_i = \left\{ p_i^x, p_i^y, p_i^f \right\} \text{ with } i = \{1, \ldots, N\},$$

built from the positions, $p^x$ and $p^y$, and the fluorescence intensities $p^f$ of each endocytic patch. The notation is similar for the reference trajectory $r$. $r$ and $p$ are already aligned together because they were acquired simultaneously.

## Temporal alignment

First, individual trajectories are aligned to their corresponding average trajectory in time. The resulting lag $\tau_p$, that aligns in time $p$ and $P$, is the one that maximizes the cross-correlation of the fluorescence intensities:

$$\tau_p = \arg \max_{\tau} \left( \sum_i p_{i+\tau}^f P_i^f \right).$$

The same procedure is used to compute the lag $\tau_r$ between $r$ and $R$.

## Spatial alignment

An isomeric transformation of space $T = \{T^x, T^y, T^\theta\}$ is the combination of a translation and a rotation

$$T : \{x, y\} \rightarrow \{\cos(T^\theta)x - \sin(T^\theta)y + T^x, \; \sin(T^\theta)x + \cos(T^\theta)y + T^y\}.$$

The optimal transformation $Tp$ that aligns $p$ to $P$ is calculated as

$$T_p = \arg\min_T \left( \frac{\sum_i w_i \left( \left( P_i^x - (Tp_{i+\tau_p})^x \right)^2 + \left( P_i^y - (Tp_{i+\tau_p})^y \right)^2 \right)}{\sum_i w_i} \right), \; \text{with } w_i = \frac{P_i^f p_{i+\tau_p}^f}{\delta_i^x \delta_i^y}.$$

The same procedure is used to compute $T_r$ that aligns $r$ to $R$. The spatial alignment of the trajectories of Abp1, Arc18 and Act1 was computed using only the trajectory data associated with the invagination dynamics, up to the trajectory peak in fluorescence intensity.

## Estimate of the average transformations

Between 50 and 350 trajectory pairs were used to compute the average alignment (*Table 1*). For each trajectory pair, we got an estimate of the transformations that align $r$ to $R$ and $p$ to $P$. An estimate of the transformation that aligns $P$ to $R$ is thus the combination of the inverse transformation that aligns $p$ to $P$ and of the transformation that aligns $r$ to $R$:

$$T = T_r (T_p)^{-1}.$$

The average transformation is then the average of the estimates of the individual transformations that align $P$ to $R$, computed from each of the trajectory pairs. The average time alignment $\tau$ is computed as:

$$\tau = \text{median}(\tau_r - \tau_p)$$

and $\delta\tau$ is the estimate of the standard error for the median computed as:

$$\delta\tau = \frac{1.4826 \times MAD_\tau}{\sqrt{M}}$$

$MAD_\tau$ is the median absolute deviation for $\tau$ and $M$ is the number of trajectory pairs used to compute the average transformation.

$T^x$ and $T^y$ are computed as:

$$T^x = \text{median}\left( T_r^x - \cos\left(T_r^\theta - T_p^\theta\right) T_p^x + \sin\left(T_r^\theta - T_p^\theta\right) T_p^y \right),$$
$$T^y = \text{median}\left( T_r^y - \sin\left(T_r^\theta - T_p^\theta\right) T_p^x - \cos\left(T_r^\theta - T_p^\theta\right) T_p^y \right).$$

$T^\theta$ is computed as:

$$T^\theta = \text{median}\left( T_r^\theta - T_p^\theta \right).$$

$\delta T^x$, $\delta T^y$ and $\delta T^\theta$ are computed as $\delta\tau$ using their respective MAD. Note that the approximation used to estimate $T^\theta$ is only valid if $T_r^\theta \approx T_p^\theta$. To ensure that this is the case, $R$ and $P$ are first aligned to their axis of symmetry, which is the direction of the invagination that we defined as the X-axis. As $r$ and $p$ are already aligned along the axis of invagination, because they were acquired simultaneously, the final rotation that align $P$ to $R$ is close to the identity ($T^\theta \approx 0$). Centering $P$ to its center of mass further minimize any error that might be induced by the above approximation.

The trajectory of the protein of interest aligned to the reference trajectory is thus:

$$P_i' = \left\{ (TP_i)^x, (TP_i)^y, P_i^f, \zeta_i^x, \zeta_i^y, \zeta_i^f \right\},$$

where $\zeta_i^x$ and $\zeta_i^y$ are the standard errors computed as:

$$\zeta_i^x = \sqrt{\left(\delta_i^x\right)^2 + \left(P_i^y \delta T^\theta\right)^2 + \left(\delta T^x\right)^2},$$
$$\zeta_i^y = \sqrt{\left(\delta_i^y\right)^2 + \left(P_i^x \delta T^\theta\right)^2 + \left(\delta T^y\right)^2}.$$

All plots describing the trajectories inward movement along the invagination direction report the 95% confidence intervals, computed as $1.96 \times \zeta_i^x$ and $1.96 \times \delta\tau$. See *Source code 2.*

## Two color averaging procedure used for non-motile proteins

Trajectory pairs of Myo5-GFP or Las17-GFP and Abp1-mCherry were used to generate Myo5 and Las17 average trajectories. The rotations and translations that aligned the Abp1-mCherry trajectories and Abp1-GFP average trajectory together were used to align the corresponding Myo5-GFP or Las17-GFP trajectories. Once aligned, the coordinates of the GFP trajectories were averaged together at each time point.

## Simulation of the accuracy of the two color alignment procedure

To control the accuracy of our alignment procedure we generated virtual trajectory pairs using two trajectories that we generated as a template (ground truth trajectories): one ground truth trajectory represents the reference protein while the other represents the target protein. The points of the trajectory pairs were randomly generated and were normally distributed around the ground truth trajectories with sigma $\sigma_P$ for the trajectories of the target protein and $\sigma_r$ for the trajectories of the reference protein. All trajectories were generated with the same sampling rate as the real trajectories. The virtual trajectory pairs were then processed with the same pipeline we used for the real data to align the ground truth trajectories together. As the ground truth trajectories were already aligned all transformations are expected to be 0 (*Figure 1—figure supplement 4A–D*). When we arbitrarily increased the noise used to generate both the reference and the target proteins we observed that the alignment procedure remains faithful up to ∼25 nm of noise. For higher values a small systematic shift occurs (*Figure 1—figure supplement 4C*). We then tested the robustness of the alignment procedure keeping $\sigma_r$ fixed at 19 nm while we varied $\sigma_p$. A $\sigma_r$ fixed mimics the noise in the real trajectory pairs, as the reference protein was always imaged under the same conditions. The alignment showed negligible errors (*Figure 1—figure supplement 3E–L*). Note that in the real trajectory pairs $\sigma_p$ span a range from 10 nm to 24 nm, however the algorithm was stable also for higher errors. We observed a systematic shift, of maximally ∼3 nm, that occurs when the trajectories are getting far from the reference (*Figure 1—figure supplement 3K*). This is expected as any small error in the alignment of the trajectories leads to an underestimate of the separation between the reference and the target protein. This underestimate grows larger the bigger the separation is, resulting in the trajectory of the target protein being shifted little closer to its reference than it should be. A shift of 3 nm is reached when the trajectories are 30 nm away far from the reference protein, which is the maximal distance of a trajectory from its reference in our model.

## Simulation of the robustness to systematic shifts between the two channels during two color acquisition

To test for the robustness against systematic shifts between the two channels, we shifted the reference trajectories in the virtual trajectory pairs that we generated on the computer (see 'Materials and Methods': Simulation of the accuracy of the two color alignment procedure; $\sigma_r = 16$ nm and $\sigma_r = 19$ nm), by shifts up to 150 nm along one direction. These shifts simulate different amounts of aberration between the two channels. On average, the alignment is not affected (*Figure 1—figure supplement 4A*). In fact, the trajectory pairs, as well as the endocytic events that they mimic, are oriented in all possible directions and once their average relative position is calculated, the chromatic aberration contribution is averaged out. However, the incertitude in the average position increases, as expected (*Figure 1—figure supplement 4A*). It is important to note that our pixel size correspond to 100 nm, therefore any shift larger than 50–100 nm would have been easily noticeable by eye. To assess how the alignment of the real data would be affected by a systematic color shift we also shifted the real trajectory pairs we used to align Sla2-GFP tp Abp1-GFP. Again, we induced shifts up to 150 nm to the trajectories of Abp1-mCherry to simulate different amounts of aberration between the two channels. The average position of Sla2-GFP does not significantly change but the incertitude in its average position increases (*Figure 1—figure supplement 4B–D*).

## Simulation of the accuracy of the trajectory averaging

To test the accuracy of the trajectory averaging we generated 65 virtual trajectories starting from a trajectory template (ground truth trajectory). The trajectories were generated adding noise that was normally distributed around the points of the ground truth trajectory with a standard deviation $\sigma$. We sampled values of $\sigma$ covering the range of noise encountered experimentally (between 10 nm and 20 nm). The average trajectories where then aligned in time and in space to a reference trajectory that was generated together with the ground truth trajectory. To compute the alignment we used virtual trajectory pairs generated from the ground truth trajectory and the reference trajectory as described in 'Materials and Methods: simulation of the accuracy of the two color alignment procedure', with noises $\sigma_p = 10$ nm and $\sigma_r = 19$ nm respectively. The averaging procedure and the complete alignment procedure are very robust: after the alignment the average trajectories were reproducing very closely the ground truth trajectory (*Figure 1—figure supplement 5*). As expected, the average of very noisy trajectories slightly underestimates the full length of the ground truth trajectory (*Figure 1—figure supplement 5D*).

## Quantification of the number of molecules

To quantify the protein amounts we imaged a sample containing cells from both a yeast strain expressing a fluorescently tagged protein of interest and a yeast strain expressing fluorescently tagged Nuf2, which was used as a reference to calibrate the fluorescence intensity (*Joglekar et al., 2006*). Both strains expressed the same fluorescent tag and were imaged together. For the quantification of the Nuf2 fluorescence intensity we used cells in anaphase-telophase only. Excitation light and emission light were directed through the U-MGFPHQ (Olympus, Japan). Samples were imaged as a z-stack of 21 frames, 200 nm spaced, using the Hamamatsu Orca-ER CCD camera. Each frame was excited with X-Cite 120Q lamp for 400 ms. Frames were not processed for background subtraction and the spots were quantified by quantifying the fluorescence intensity of the patches, in the frame of the z-stack in which they were brighter, and subtracting their local background, as described in *Joglekar et al. (2006)*.

The intensities of the fluorescent patches of the target endocytic proteins where in general dimmer than Nuf2, and their distribution was in general not symmetric. The intensities measured from the patches of the target proteins were thus processed after a logarithmic transformation. The average number of molecules of the target protein $n_p$ was derived as:

$$n_p = \frac{f}{g} n_r.$$

$n_r$ is the known number of molecules of the reference, $f$ is the median of the fluorescence intensity of the patches of the target protein and $g$ is the median fluorescence intensity of the patches of the reference protein. The uncertainty in the number of molecules was derived as:

$$\delta n_p = \sqrt{\left(n_r \frac{f}{g} \delta_l\right)^2 + \left(n_r \frac{f}{g^2} \delta_g\right)^2 + \left(\frac{f}{g} \delta n_r\right)^2}. \tag{1}$$

$\delta_l$ is the estimate of the standard error for the median of the fluorescence intensity of the target protein after logarithmic transformation, $l = \log(f)$, and is computed as:

$$\delta_l = \frac{1.4826 \times MAD_l}{\sqrt{N}}.$$

$MAD_l$ is the median absolute deviation for $l$ and $N$ is the number of observations. The distribution of the intensities of the target endocytic proteins where not symmetric, they were thus processed after a logarithmic transformation of their intensities. $\delta_g$ is computed as $\delta_l$, using $MAD_g$, the median absolute deviation of the fluorescence intensity of the reference protein. $\delta n_r$ is the uncertainty in Nuf2 number of molecules. The average number of molecules $n_p$ was then used to rescale fluoresce intensity curves of the average trajectories (see 'Materials and Methods': calibration of the fluorescence intensity curve of the trajectories with the number of molecules).

Nuf2 number of molecules was quantified using Cse4 as a reference. The number of Cse4 molecules used for the calibration was 5 molecules/kinetochore (*Lawrimore et al., 2011*). The measured average number of Nuf2 molecules, per fluorescent spot, was $280.6 \pm 16.1$ molecules. Its error was quantified as

in *Eq.(1)* with $\delta n_r = 0$. The quantification of Nuf2 molecules with Cse4 served as a control for our procedure as there is no difference in the ratio between our number of Nuf2 molecules and the number of Cse4 molecules, which is $3.5 \pm 0.2$ Nuf2 molecules each Cse4, and Joglekar's ratio, which is 3.5.

To quantify the protein abundance of Arc18, which was tagged with myEGFP, we compared the fluorescence intensity of myEGFP and EGFP tags and we measured the myEGFP tags to be $68\% \pm 14\%$ of the fluorescence intensity of an EGFP tag. The uncertainty in Arc18 number of molecules was then computed as:

$$\delta n_p = \sqrt{\left(c\frac{f}{g}n_r\delta_l\right)^2 + \left(c\frac{f}{g^2}\delta_g\right)^2 + \left(c\frac{f}{g}\delta n_r\right)^2 + \left(\frac{f}{g}n_r\delta_c\right)^2}.$$

$c$ is the estimate of the correction for the fluorescence intensity and $\delta_c$ is its uncertainty.

## Calibration of the fluorescence intensity curve of the trajectories with the number of molecules

We calibrated the fluorescence intensity curve $P^f$ for each average trajectory,

$$P_i = \left\{P_i^x, P_i^y, P_i^f, \delta_i^x, \delta_i^y, \delta_i^f\right\} \text{ with } i = \{1, \ldots, N\},$$

to estimate of the number of molecules $P^n$ over time. $P^n$ is computed by rescaling the fluorescence intensity curve of the protein of interest with the average number of molecules $n_p$ at the endocytic site (see 'Materials and methods': quantification of the number of molecules):

$$P_i^n = n_P \frac{P_i^f - P_{min}^f}{\frac{1}{N}\sum_{j=1}^{N}\left(P_j^f - P_{min}^f\right)}, \text{ with } P_{min}^f = \min_k\left(P_k^f\right)$$

The error in the estimate of the number of molecules is:

$$\delta_i^n = \sqrt{\left(\frac{P_i^f - P_{min}^f}{\bar{F}}\delta n_P\right)^2 + \left(\frac{n_p}{N}\frac{N\bar{F} - P_i^f + P_{min}^f}{\bar{F}^2}\delta_i^f\right)^2 + \left(n_p\frac{P_i^f - P_{min}^f - \bar{F}}{\bar{F}^2}\delta m\right)^2},$$

$$\delta m = \delta_l^f \text{ with } l = \arg\min_j\left(P_j^f\right),$$

$$\bar{F} = \frac{1}{N}\sum_{i=1}^{N}\left(P_i^f - P_{min}^f\right).$$

where $\delta n_p$ is the standard error of the average number of molecules measured in the endocytic fluorescent patches (see 'Materials and Methods: quantification of the number of molecules). All plots describing the number of molecules report the 95% confidence interval, computed as $1.96 \times \delta_i^n$.

## Quantification of the ratio between GFP-Act1 and the total actin

To quantify the ratio between GFP-Act1 and the total actin we run western blots with cell extract of yeast cells expressing GFP-Act1 (MKY2653) (*Wu and Pollard, 2005*). MKY2653 cells where grown overnight in SC-URA media. As primary antibody against actin we used Sigma A2066. The secondary antibody was a AP-1000 Alkaline Phosphatase anti-rabbit igG (H + L) from Vector Laboratories. The quantification was repeated 8 times. The ratio $\tilde{r}$ between GFP-Act1 and the total actin was computed as:

$$\tilde{r} = \frac{r}{r+1}$$

where $r$ is the average ratio between the GFP-Act1 and the endogenous actin that we measured from the 8 quantifications. The error was computed as:

$$\sigma_{\tilde{r}} = \frac{\sigma_r}{(r+1)^2}$$

where $\sigma_r$ is the standard error of the mean of $r$.

## The direction of Abp1 trajectories with respect to the membrane

To determine the angle between Abp1 trajectories and the yeast cell surface, we determined the closest membrane tangent for each Abp1 track, using a binary mask of the yeast cell in which the trajectory was acquired. We then measured the angle between the vector tangent to the membrane and the vector whose direction was determined by the interpolation of Abp1 trajectory points on the focal plane (*Figure 1—figure supplement 2B*).

## Alignment of photobleaching trajectories

Individual photobleaching experiments were tracked with the Particle Tracker (*Sbalzarini and Koumoutsakos, 2005*) plugin in ImageJ after background subtraction, normalization and cytoplasmatic background subtraction of the images (see 'Materials and methods': Image analysis). The tangent to the plasma membrane was determined in ImageJ using a binary mask of the cell. A custom written software in R was then used to extrapolate the trajectory along the direction orthogonal to the plasma membrane, which represents the inward movement of the photobleached endocytic spot. The resulting trajectories were aligned in time and in space to the average trajectory of the corresponding protein using the fluorescence intensity curve and the inward movement of the average trajectory and of the photobleached trajectory before the photobleaching. After alignment, the photobleached trajectories were thus aligned in space and time with all the average trajectories and with the plasma membrane profiles (*Figure 6C–F*).

## Measuring the displacement of N- and C-termini of Sla2

To check whether Sla2 was oriented on average perpendicularly to the plasma membrane, we tagged simultaneously the N-terminus with sfGFP and the C-terminus with mCherry (GFP-Sla2-RFP, *Figure 3—figure supplement 1A*). In cells treated with 2 μM LatA, we recorded the closest membrane tangent to each GFP spot. We then measured the angle between the vector tangent to the membrane and the vector whose direction was determined by the centroids of the corresponding GFP and mCherry spots, for each GFP and mCherry pairs (*Figure 3—figure supplement 1B*).

In order to determine whether the displacement between the N- and C-terminal trajectories of Sla2 is a measure for Sla2 length we measured the distance between the N- and C-terminus of Sla2, by tagging simultaneously the N- terminus with GFP and the C-terminus with mCherry (*Figure 3—figure supplement 1A*). We arrested the membrane invagination by treating cells with 2 μM of LatrunculinA (LatA) (*Kukulski et al., 2012*) and we imaged cells on the GFP and mCherry channels. Images were corrected for chromatic aberration as for the two color trajectories (see 'Materials and methods': Two color alignment procedure). The separations between the centroids of the GFP and mCherry pairs follow a non-gaussian distribution (*Stirling Churchman et al., 2006*), which was used to compute the distance between the fluorophores and thus estimate the displacement of the N- and C-terminal tags of Sla2 (*Figure 3—figure supplement 1C*). The distance is reported in the text together with the estimate of the standard error of the mean obtained from the observed Fisher information matrix. As a control, we performed the same analysis on a TetraSpeck sample (*Figure 3—figure supplement 1D*) and on Sla2 tandemly tagged at its C-terminus with both mCherry and sfGFP (*Figure 3—figure supplement 1E,F*).

## Estimation of the membrane area covered by Rvs

To estimate the membrane area covered by the Rvs161/167 proteins, we assumed that all protein molecules are membrane bound and homogeneously distributed on the surface of the endocytic invagination. Their distribution was computed considering that BAR proteins form a dimer that is ~13 nm long (*Peter et al., 2004*) and tubulate in vitro forming spirals spaced by 50 Å (*Mim et al., 2012*). We used those data, together with our estimate of the number of Rvs molecules and with the average membrane shapes, to model all possible coverages of the invagination along the invagination length. We then chose the coverage whose center of mass matched the position of the Rvs167-GFP average trajectory. We extrapolated the plasma membrane profiles to estimate the plasma membrane shape at time 0 in order to determine the Rvs coverage just prior to scission.

## Estimation of Abp1 patch lifetime (*Figure 1—figure supplement 2A*)

Abp1 lifetimes were estimated from the lifetime of the trajectories (i.e. the time difference between the first and the last time point in the trajectory). We considered only the trajectories that were complete and, to minimize as much as possible the effect of photobleaching, we considered only the trajectories that were recorded in a number of frames, at the beginning of the video, covering ~3 times the known lifetime

of the protein. Note that to achieve robust automatic patch detection the thresholding of the fluorescent patches was stringent, resulting in Abp1 patch lifetimes that are an underestimate of the true patch lifetime.

## Sample preparation for superresolution microscopy

ConA crosslinked glass coverslips were prepared as described previously (*Mund et al., 2014*). 24 mm coverslips were cleaned for at least 12 hr in 1:1 methanol/hydrochloric acid, and washed in ddH$_2$O until the pH of the solution remained neutral. 20 µl of Bioconext (UCT, Bristol, PA) was spread out over the coverslip and incubated for 30 min, followed by washing twice with ethanol, twice with H$_2$O, drying at 65°C for 30 min and incubation for 60 min with 20 µl of 2% ConA. Coverslips were then rinsed three times with H$_2$O, air dried and stored until usage.

Sample preparation was performed as described previously (*Mund et al., 2014*). In summary, yeast cells were grown in SC-Trp medium until reaching log phase, pelleted by centrifugation, resuspended in a small volume of ddH$_2$O and pipetted on the ConA crosslinked glass coverslips. After settling for 15 min, the supernatant was removed and the coverslips were submerged in a fixative containing 4% formaldehyde, 2% sucrose in PBS for 15 min. Cells were then incubated two times for 15 min in PBS containing 50 mM NH$_4$Cl to stop the fixation.

Coverslips were subsequently put face down on a 100 µl drop of blocking solution consisting of 50% ImageIT FX (Invitrogen, Carlsbad, CA) to block background due to unspecific binding and 0.25% Triton-100 in PBS for 60. Coverslips were briefly washed three times with PBS and put face down on a 100 µl drop of SNAP labeling solution containing 1 µM SNAP-Surface Alexa Fluor 647, 1% BSA, 0.25% Triton X-100, 0.004% NaN$_3$ in PBS and incubated for 120 min.

Finally, coverslips were washed three times by gentle shaking in PBS for at least 5 min.

## Superresolution microscopy

Localization microscopy was performed on a custom-built microscope. Single-mode output from an iChrome MLE-L laser box equipped with 405 nm, 488 nm, 561 nm and 640 nm laser lines (Toptica Photonics, Germany) was focused onto the back–focal plane of a 60× NA 1.49 TIRF objective (Nikon, Japan) and adjusted for epi illumination. Emission light was filtered using an ET 700/100 bandpass filter (Chroma, Bellows Falls, VT) and a FF01-446/523/600/677 multi bandpass filter (Semrock, Rochester, NY), and focused by a 400 mm tube lens onto the chip of an EMCCD camera (Ixon Ultra, Andor, United Kingdom) that was air-cooled to −75°C. Images were acquired using MicroManager (*Edelstein et al., 2010*).

A piezo objective positioner (Physikinstrumente, Karlsruhe, Germany) was used to move the z-focus. The focus was stabilized by an electronic feedback loop based on an infrared laser that was totally internally reflected at the coverslip and detected by a quadrant photodiode. The z stability was better than ±10 nm over several hours. Lateral drift, typically smaller than 50 nm/hr, was corrected for in the analysis software.

Image acquisition was performed as described (*Mund et al., 2014*). In brief, the samples were mounted in a custom-made holder and covered with at least 200 µl of imaging buffer consisting of 150 mM Tris–HCl pH7.5, 2% glucose, 60 mM cysteamine, 40 µg/ml catalase and 0.5 mg/ml glucose oxidase. We used an exposure time of 25 ms and an EM gain of 100. Imaging laser intensity at 640 nm was 2.5 kW/cm$^2$, the activation laser intensity was automatically adjusted to ensure a constant number of localizations per frame. Typically 30,000–50,000 frames were recorded.

Localization analysis was performed as previously described (*Ries et al., 2012*). In summary, photon counts were obtained by subtraction of the constant offset from the pixel count and multiplication with the inverse gain. Initially, approximate positions of bright spots were determined by smoothing, nonmaximum suppression and thresholding. Selected peaks were fitted by a pixelized Gaussian function and a homogenous photonic background with a maximum likelihood estimator for Poisson distributed data using a freely available, GPU based fitting routine (Smith et al., 2010) on a Geforce GTX670 (Nvidia, Santa Clara, CA). Lateral drift correction was performed using image correlation as previously described. Localizations with an uncertainty of >15 nm were discarded. The images in *Figure 3* and *Figure 3—figure supplement 2* were rendered using a Gaussian with a width according to the respective localization precision. All analysis software was written in Matlab (TheMathworks, Natick, MA).

The observed structures in *Figure 3—figure supplement 2* were visually classified. 'Clear ring structures' show both localizations in circular arrangement and a pore zone in the middle with virtually no localizations, while for 'Possible ring structures' one of the two criteria was less striking. 'No ring structures' exhibit other geometries.

## Acknowledgements

The authors thank M Knop and M Meurer for plasmids and the yeast strain expressing Sla2 C-terminal tandem fluorescent protein tag (tFT-tag), O Gallego for the strain MKY2119 and for discussion and help in designing experiments for Sla2 length estimation, C Häring for antibodies, W Kukulski, M Schorb, T Brach, Y Frosi, C Godlee, B Klaus (Center for Statistical Data Analysis) and K Miura (Center for Molecular and Cellular Imaging) for discussions. AP acknowledges an EIPOD fellowship and the Deutsche Forschungsgemeinschaft (DFG, KA 3022/1-1).

## Additional information

### Funding

| Funder | Grant reference number | Author |
|---|---|---|
| EMBL Interdisciplinary Postdoc fellowship | | Andrea Picco |
| Deutsche Forschungsgemeinschaft (DFG) | KA 3022/1-1 | Andrea Picco, Marko Kaksonen |

The funders had no role in study design, data collection and interpretation, or the decision to submit the work for publication.

### Author contributions

AP, MM, Conception and design, Acquisition of data, Analysis and interpretation of data, Drafting or revising the article; JR, FN, MK, Conception and design, Analysis and interpretation of data, Drafting or revising the article

### Author ORCIDs

Andrea Picco, http://orcid.org/0000-0003-2548-9183
Markus Mund, http://orcid.org/0000-0001-6449-743X
Jonas Ries, http://orcid.org/0000-0002-6640-9250
François Nédélec, http://orcid.org/0000-0002-8141-5288
Marko Kaksonen, http://orcid.org/0000-0003-3645-7689

## Additional files

### Supplementary files

• Supplementary file 1. Average trajectory data. The average trajectories plotted in *Figures 1–6* **t**: time points in seconds (time 0 corresponds to the Rvs167 intensity peak). **t.err**: 95% confidence interval of the time alignment in seconds. **x**: centroid position, in respect to the plasma membrane and along the invagination axis (nm). **x.err**: 95% confidence interval of the centroid position, in respect to the plasma membrane and along the invagination axis (nm). **n**: estimate of number of molecules. **n.err**: 95% confidence interval of the number of molecules.

• Source code 1. Average trajectories V0.1. The collection of R functions used to compute the average trajectories. They require the R library Hmisc.

• Source code 2. Align trajectories V0.2. The collection of R functions used to align the average trajectories together and to plot them. They require the R library MASS and the function plotCI form the R package gregmisc.

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
