## [Decision Letter]

Thank you for sending your work entitled “Visualizing the functional architecture of the endocytic machinery” for consideration at *eLife*. Your article has been favorably evaluated by Vivek Malhotra (Senior editor), a Reviewing editor, and 3 reviewers.

The Reviewing editor and the reviewers discussed their comments before we reached this decision. In summary, all the reviewers agree that you have taken the analysis of endocytosis to a new level. However, presentation in *eLife* requires that the work also advance our understanding of an important biological process. We would be happy to reconsider the work if it includes an advance in our knowledge, in addition to addressing the detailed comments that follow below. Please pay particular attention to the critique on the method description, accuracy and tests. (Because the reviews are lengthy, we include them all to help you.)

Reviewer #1

This manuscript presents a comprehensive analysis of the dynamics of components of the Clathrin-mediated endocytosis machinery in yeast *Saccharomyces cerevisiae*. The authors used a centroid-based estimation of localization. Since the number of molecules on the clathrin coated vesicles CCV is in the range of 50-100 and the CCV have a regular shape, the center of mass of molecules can be localized with a few nm accuracy. The authors used mCherry labeled Abp1 as a base line and aligned other components of the endocytic machinery relative to Abp1 in space and time. This approach yielded a comprehensive description of the primary events occurring during CME in yeast.

The idea of aligning proteins to the center of mass of other proteins and building a comprehensive description of the relative activity of the endocytic machinery in space and time is interesting and promising. Since the method provides the time resolved behavior of the center of mass, the more detailed spatial organization of molecules on CCV was studied by CLEM (correlation of light microscopy with EM) and super-resolution microscopy. Using this approach, the authors describe the order of assembly and movement of the CCV proteins during invagination. The FRAP experiments allowed them to discriminate between different models of actin polymerization.

The study is potentially interesting but there are two main problems, one is technical and the other conceptual. The accuracy of the results critically depends on the accuracy of the trajectory alignment procedure used. Unfortunately, the description of the alignment procedure is cryptic (see below). In addition, even with an accurate procedure, every computer-based procedure will in all likelihood contain some implementation bugs. Therefore, its accuracy has to be tested in well controlled experiments.

One additional problem is that the precise question addressed by this study is unclear. I do not wish to dispute the fact that the quantitative analysis of CCV assembly and invagination is an important contribution. However, such a thorough analysis should also translate into some significant conceptual advances to merit publication in *eLife*. This is not apparent from this study. There are a number of potentially interesting points that the authors make, such as the specific orientation of coat-associated proteins, the order of proteins (e.g. Sla1 and End3) assembly, the formation of a BAR domain lattice on the tubular part of the invagination, the predicted bending forces exerted on the membrane, the activation and termination of actin filaments nucleation prior to and following vesicle scission. My impression however is that these are only incremental advances more than a true discovery.

Technical comments:

1) The accuracy of correction of chromatic aberration was not estimated, but it is expected to be in the range of measurable values. The alignment to the Abp1-mCherry is valid, because the systematic shift is common for the same type of fluorescent protein. However, the comparisons of shifts are valid only between the same fluorophores, e.g., Sla1-GFP to End3-GFP, but they are not for GFP and mCherry without a quantification of chromatic correction uncertainty.

2) This work is critically dependent on the accuracy of trajectory alignment. Unfortunately the procedure of two color alignment is very hard to follow. For example, in the Materials and methods: “These trajectory pairs were then used to align the average trajectory of the protein of interest to the average trajectory of Abp1-GFP […] that aligned the Abp1-GFP average trajectory to the Abp1-mCherry trajectory in the trajectory pair.” I am confused about which average trajectory aligns to which and how they are averaged. Were the tracks of Abp1-mCherry and of the GFP-proteins aligned independently of each other? I do not find it logical that a two-step alignment has higher accuracy than a single step alignment, against what the authors claim. All additional procedures have to increase uncertainty. In the case of doubling the number of measurements, the uncertainty decreases only by a factor square root of 2. Without a detailed description and controlled test it is impossible to judge the real accuracy of the proposed procedures. The correctness of error propagation is critical for the conclusions, but it is not reported in detail.

3) The number of molecules was estimated by a two-step comparison of fluorescence of the kinetochore proteins Nuf2 and Cse4. Unfortunately, the authors did not estimate the increase of the uncertainty by the additional step, but just multiplied the uncertainty of the first step by a scale factor (see Methods “Quantification of the number of molecules”). As such, the accuracy of number of molecules is overstated.

4) In the chapter “Assembly of Rvs proteins at the neck of the invagination” the authors described fitting procedures and estimations that are supposed to be numerical. Unfortunately, they did not provide any quantitative data and fit uncertainty estimation, but only a cartoon on Figure 4. The small Figure 4 just gives a crude value 100+-(?) molecules in the scission neck. The curve on Figure 4 raises the question of whether a fraction of Rvs has to be on the plasma membrane side of the scission neck. After cutting the neck the membranes with Rvs move to opposite directions, with the vesicular part moving towards the interior and the PM part moving retrogradely. One could expect that it will be reflected in an oscillatory behavior of centroid with mean position in the center of the neck and high level of uncertainty. Unfortunately, this process is below the time resolution of the authors' technique. The curve on Figure 4 shows a fast shift toward the vesicle. The authors commented that Rvs persists 2 sec longer on the vesicle than the plasma membrane. But another possible explanation is that a larger fraction of Rvs localized above the scission plane, something that would be inconsistent with the cartoon on Figure 4. The spreads of red and green points on Figure 6 are clearly different, but the difference in SD is rather minor (15 and 18nm).

Figure 1—figure supplement 2: units of Y-axis are neither on the figure nor in the legend.

Figure 1—figure supplement 2 has 3 mistakes:

1) The optical slice depth 400nm is an overestimation. A well aligned confocal microscope has f ∼ 750nm.

2) I suppose that the radius of a cell was taken equal 2.5 um, but it was not mentioned on the figure and in the legend.

3) In the formula on 2D there is a mistake: the square root is missing. It has to be sqrt(1-f^2/(4 * r ^2)) * I. The estimation of maximal error is done with an over-optimistic optical depth and by a wrong formula that result in a 0.6% value in the legend. Moreover, this value is inconsistent with the value in the main text where it was given as 10%.

On Figure 5 the color scheme is far from optimal. I could not figure out from the figure and legend which tracks belong to Abp1 and which to Myo5.

Video 2 is packed with some uncommon codec, so I could not see it on two computers under 64bit Win7 and 32bit Win8 OS respectively.

Reviewer #2

Major novel findings are limited as no new concepts are discovered. The idea that actin polymerizes primarily at the base of the pits is not really new (due in part to work of Kaksonen himself; a google search for images of actin at yeast actin patches or clathrin coated pits reveals that most schematic cartoon place actin polymerization at the base of the invagination), although this is proved elegantly here. On the other hand, this study is highly quantitative, fluorescent proteins are expressed at endogenous levels and the findings are supported by a large number of controls.

Issues to be addressed with discussion or new experimentation:

A major open question in the field is whether the dynamin homologue, Vps1, participates in fission at yeast actin patches. The authors mention Vps1, but do not show any data. Given the focus of this study, it would be great if the authors could analyze Vps1 and make a conclusive statement on this highly controversial topic.

The experiments with photobleaching of actin do not go past the time of scission (time = 0s). Thus, it remains unclear whether actin polymerization continues after scission. In this connection, several studies carried out on mammalian cells have demonstrated the occurrence of actin tails propelling newly formed endocytic vesicles. While tails were first described at sites of non-clathrin-dependent endocytosis (Merriefield and Almers, PMID: 10559868), a recent *eLife* paper demonstrated that also clathrin coated vesicles nucleate actin tails when the dephosphorylation of PI(4,5)P2 is impaired (Nandez et al., PMID: 25107275). Thus, while the present study proves what had been implicit in previous studies, i.e. that actin nucleation occurs at the base of the invagination, could some actin eventually nucleate also from the newly formed vesicle (for example at the site of the scar left by the fission)? The authors show that a pool of Rvs remains partially associated with the vesicle and one wonders whether some actin nucleating factors may be present at the same site.

Concerning Rvs (the endophilin homologue): it was shown that endophilin deficiency at neuronal synapses results in a defect in uncoating, most likely due to a deficient recruitment of synaptojanin, a PI(4,5)P2 phosphatase (Milosevic et al., PMID: 22099461). The Rvs pool that remains associated with the endocytic vesicles may be needed for uncoating, as Rvs167 binds a yeast synaptojanin. The authors may want to comment on this.

The authors state that as the number of Las17 molecules start decreasing before fission, suggesting that fission may not be assisted by the force generated by actin polymerization. Yet, one cannot exclude that the remaining Las17 may still be needed to generate a force. Photobleaching experiments show continued actin nucleation at least until the time of fission. As tagged Las17 does not behave as wild type Las17, these experiments must be interpreted with caution.

Why is Las17 not drawn bound to the membrane in the cartoon of Figure 7? Las17/nWASP are actin nucleators at the plasma membrane.

The authors state that the number of Sla2 molecules remains relatively constant prior to scission, yet this is contrary to the data provided (Figures 2, 3 and 7), which shows that the number of Sla2 molecules starts declining a few seconds before scission.

Despite the thorough analysis of the correlation between changes in membrane shape and the recruitment of endocytic components, it remains unclear what causes the initial buckling of the plasma membrane. This work favors the idea that the initial deformation coincides with the nucleation of actin (Figure 7), in line with the view presented in Kukulski, et al. (2012). The stronger evidence in favor of this model is that in the presence of Latrunculin A, no invaginations are observed (40). [30] however, shows that there is an initial deformation of the membrane even in the presence of Latrunculin A. An alternative model suggests that early endocytic proteins like Syp1, Ede1 and Ent1 may act as the initiators of membrane curvature. What does the recruitment of these early endocytic proteins look like in this model and how does it relate to changes in membrane shape? Is the initial deformation the result of early endocytic proteins like Syp1, Ede1 and Ent1 coupled with actin polymerization? I note that, as often pointed out by Tom Kirchhausen who has extensively studied the structure of the clathrin lattice, it is unlikely that a flat clathrin lattice may convert to a curved clathrin lattice without undergoing a complete disassembly. Thus, it would seem more likely that the initial coated patch may already be curved. I realize that a thorough analysis of the spatio-temporal localization of early endocytic factors may go beyond the goal of this study, but these considerations should be discussed.

Reviewer #3

This is rigorous, technically demanding work on an interesting biological process. I am generally enthusiastic about the work and the presentation, but think that the authors should have taken a broader perspective on their work, since their work confirms many of findings and ideas of others working on fission yeast. That previous work does not detract from the value of the present work, but it should not be ignored.

Scientific issues:

Results section: “We thus produced a dataset in which all average trajectories are aligned to Abp1.” I am not concerned that some information was lost during averaging. The individual tracks in Figure 1 average out all of the lateral motion of the patches, because the motions to the right and left of the normal (zero on the x-axis) to the plasma membrane cancel out. This representation may not give the casual reader the right impression about the patch motions, which would be more apparent with plots of the averages of the absolute values of the left and right positions.

In the first paragraphs of the Results section: It might be informative to compare the patch alignment method used here with that of Berro (MBoC, 2014). My impression is that the two methods work about equally well. Berro discusses the biological variability. Do you have any measure of the biological variability in your system, since you seem to assume none. You should state this assumption, if you made it.

In the subsection “Organization of coat associated proteins”, in Results: An average localization precision of approximately 10 nm is extremely good; what is the evidence that the resolution is actually 10 nm?

In the beginning of the subsection “Assembly of Rvs proteins at the neck of the invagination”, in Results: The approach to calculating the membrane area covered by the Rvs dimers is identical to Arasada (2011) in fission yeast. Comparing the results may be informative.

In the same section: Berro, MBoC (2014) has evidence that the movements of the actin-covered vesicle are diffusive in the cytoplasm.

In the beginning of the subsection “Assembly of the actin cytoskeleton”: “The Myo5 trajectory remained almost stationary during the invagination of the plasma membrane, whereas the Las17 trajectory moved inward, but much less than Abp1, Arc18 and Act1 trajectories.” The behavior is similar in fission yeast (Arasada, 2011).

In the same section of the text, you state: “Note that because GFP-Act1 was expressed in addition to untagged actin, we do not know the ratio between tagged and untagged molecules that are recruited to the endocytic site and we can only provide relative abundance estimates.” Rather than using arbitrary units for Act1, measure the ratio of GFP-Act1 and wild type Act1with an anti-actin immunoblot and give the numbers of GFP-Act1 and the fraction tagged, allowing for a calculation of the total actin. Wu (2005) and Sirotkin (2010) explain how to do this.

In the beginning of the subsection “Actin polymerizes at the base of the invagination”, in Results: The FRAP experiments are elegant and informative. Abp1 is a nice control. However the authors do not include the most obvious model in Figure 6, namely that new branches are formed on pre-existing filaments in proportion to the local concentration of the NPFs and then grow in random directions. This is what is expected from the mechanism of branch formation and does not presuppose a mechanism to align the branches in a certain direction (which seems unlikely). Arasada (2011) uses this assumption and the data in Figure 6 strongly support it. My impression is that the FRAP data are consistent with this model with local nucleation and random elongation, which the authors could simulate knowing the local concentrations of the two NPFs as established in this paper. Capping may be favored near the membrane (as assumed by some models of leading edge motility) and may bias the direction of growth, but the authors do not consider capping or other mechanisms to bias growth.

In the subsection headed “The endocytic coat”, in the Discussion, you state: “The disassembly of Sla1 and End3 begins after membrane invagination has started, but several seconds before scission, suggesting that these proteins may be important during early stages of vesicle budding.” This supports the model of Chen (2013) with adaptor proteins binding actin filaments to provide mother filaments to start the branching process.

Still in the Discussion, the first paragraph of the subheading “Endocytic actin network”: Berro (2010) uses data similar to that in this paper to calculate the evolution of the rates of the main actin assembly reactions from the beginning to the end of the patch life, so one does not need to speculate about “these data suggest that the nucleation of new actin filaments stops before vesicle scission occurs.” Dissociation of the NPF's is enough to explain the slowing of the rate of branching.

Subsection headed “The functional organization of the endocytic machinery”: Figure 7 is the budding yeast version of fission yeast data in Figure 7 and Table 1 in [63]. Arasada 2011 and Chen 2013 have additional quantitative data on other fission yeast actin patch proteins. The timing and numbers of molecules are remarkably similar in the two yeast in spite of >400 my of divergent evolution. This certainly deserves some discussion. The new data in this paper leads to conclusions similar to those in the fission yeast papers. Given this new data from budding yeast, it would be interesting to make a formal comparison with fission yeast actin patches, including the peak times, peak numbers and depth of penetration of each homologous molecule: actin, Arp2/3 complex, Wsp1 (Las17), Myo1 (Myo5), Cdc15 and Bzz1 (Rvs167) and End4 (Sla2).

Figure 7 legend: I would call the GFP-actin signal “polymerized actin” rather the “actin cytoskeleton,” since you use the latter term for actin filaments plus associated proteins. Since you can calculate the total numbers of polymerized actin molecules at each point along these curves and since you know the numbers of Arp2/3 complex, you should be able to make some simple calculations such as the lengths of the filaments and numbers of branches formed each second.

Figure 7 legend: “Time ≈ -8 s: actin polymerization starts.” This drawing is misleading, since it shows Sla2 binding a branch, whereas the biochemical evidence strongly shows that branching only occurs on the side of a pre-existing filament. Where do the pre-existing filaments come from? Chen (2013) has one possible source. The drawings of the later time points show that the network propagates correctly from mother filament to branch, but this is not obvious, so it would be helpful to have some clue showing that the first branches end up at the tip of the invagination.

The model considers the branches to be stable, but actually the branches and the filaments are actually turning over rapidly on the time scale of this figure (as shown by the simulations of Berro, 2010). It would be helpful if the figure or the text could indicate this turnover.

Note that Berro, MBoC, 2014, has more realistic scale drawings the filaments in slices through actin patches. If the actin numbers are similar in budding yeast actin patches, you could make less schematic drawing of the actin network base on his calculations.

[Editors' note: further revisions were requested prior to acceptance, as described below.]

Thank you for resubmitting your work entitled “Visualizing the functional architecture of the endocytic machinery” for further consideration at *eLife*. You will be pleased to learn that your revised article has been favorably evaluated by Vivek Malhotra (Senior editor) and Suzanne Pfeffer, a member of the Board of Reviewing Editors, and two reviewers. The manuscript has been improved but there are some remaining issues that need to be addressed before acceptance.

The referees were pleased that you took the reviewers' suggestions seriously. Nevertheless, Thomas Pollard (peer review) wrote: ”My main concern with the original paper was its narrow point of view. The presentation generally ignored highly relevant work on other organisms, especially fission yeast. The new work on budding yeast offers valuable, new insights, but much of the story is remarkably similar to what was found in fission yeast. Making this clear highlights the amazing conservation of the mechanisms over the hundreds of millions of years since these two yeast diverged. These comparisons detract in no way from the authors' accomplishments. Also, the authors seem to have misunderstood several suggestions to compare the data from the two yeasts. They declined to do so, saying that this would require new experiments. However, all they needed to do was to plot the published data on the same graphs as their new data and comment on what is the same and what is different.

Second, citations: The authors added citations to some of the work on fission yeast, but missed some opportunities to inform a reader about what was known. Here are two examples:

Counting total actin from the numbers of GFP-actin and the ratio of GFP-actin to untagged actin: the new data look good, but the method developed by Wu was not cited.

Idea that adaptor proteins bind actin filaments to provide mother filaments for Arp2/3 complex to start the branching process: The revised text makes this point and cites four papers, three dealing with the discovery of the adapter proteins, none of which mention Arp2/3 complex. This gives the impression that these early paper rather than the Chen paper came up with the idea.

Finally, (not essential) in terms of the idea that the dissociation of the NPF's is enough to explain the slowing of the rate of branching: I was hoping that they would use their data to calculate the rates of the reactions, as done before by others. This would be nice but is not essential.”

---

## [Author Response]

Reviewer #1

*[…] The study is potentially interesting but there are two main problems, one is technical and the other conceptual. The accuracy of the results critically depends on the accuracy of the trajectory alignment procedure used. Unfortunately, the description of the alignment procedure is cryptic (see below). In addition, even with an accurate procedure, every computer-based procedure will in all likelihood contain some implementation bugs. Therefore, its accuracy has to be tested in well controlled experiments*.

*One additional problem is that the precise question addressed by this study is unclear. I do not wish to dispute the fact that the quantitative analysis of CCV assembly and invagination is an important contribution. However, such a thorough analysis should also translate into some significant conceptual advances to merit publication in eLife. This is not apparent from this study. There are a number of potentially interesting points that the authors make, such as the specific orientation of coat-associated proteins, the order of proteins (e.g. Sla1 and End3) assembly, the formation of a BAR domain lattice on the tubular part of the invagination, the predicted bending forces exerted on the membrane, the activation and termination of actin filaments nucleation prior to and following vesicle scission. My impression however is that these are only incremental advances more than a true discovery*.

The reviewer brings out two issues: a technical problem, the lack of a detailed description of the method and its accuracy, and a conceptual problem, the lack of a true discovery.

We agree that the description of the method was rather cryptic and we have written a clearer and more detailed description of the alignment procedure. In addition, as suggested we performed extensive simulations that test the robustness of our method and the correctness of the software. These tests prove the accuracy of our method (Figure 1—figure supplement 3, Figure 1—figure supplement 4 and Figure 1—figure supplement 5). The accuracy of our method is further supported by the experimental controls: the alignment of Abp1-GFP to itself (Figure 1—figure supplement 2) and the separation between GFP-Sla2 and Sla2-GFP (Figure 3—figure supplement 1).

We disagree with the conceptual concern raised by the reviewer. Our work presents a number of new discoveries that are critical for a mechanistic understanding of the endocytic machinery: we show how key coat components (Sla2, End3 and Sla1) are spatially organized to perform their specific tasks in vesicle budding. We describe a quantitative dynamic model for the coverage of the plasma membrane invagination by N-BAR proteins in vivo. Finally, we solve the long lasting controversy about the location of actin filament polymerization. Understanding the spatiotemporal organization of actin polymerization is essential for understanding the mechanism of actin driven endocytosis.

Furthermore, we are absolutely convinced that for the mechanistic understanding of endocytosis (or any other cellular processes) a systematic quantitative analysis of its components in vivo is essential. We therefore believe that our results, which describe at the nm scale the interplay between the endocytic protein machinery and the membrane, should be considered “true discoveries”.

*Technical comments*:

*1) The accuracy of correction of chromatic aberration was not estimated, but it is expected to be in the range of measurable values. The alignment to the Abp1-mCherry is valid, because the systematic shift is common for the same type of fluorescent protein. However, the comparisons of shifts are valid only between the same fluorophores, e.g., Sla1-GFP to End3-GFP, but they are not for GFP and mCherry without a quantification of chromatic correction uncertainty*.

We have added simulations and controls to demonstrate how the quality of the alignment depends on the aberration error. There are two sources of color related aberration in our set-up: the chromatic aberration and the aberration in the imaging setup used for the simultaneous acquisition of the two channels. We correct for both types of errors simultaneously. Our new simulations demonstrate that a systematic color shift between the centroid positions, does not significantly affect our trajectory alignment (See Materials and methods: Simulation of the accuracy of the two color alignment procedure; Figure 1—figure supplement 4). Interestingly, a systematic color shift between mCherry and GFP does not affect the relative positions of the average trajectories after the alignment, only the uncertainty in their position (Figure 1—figure supplement 4). The reason is that the endocytic events in cells are oriented randomly within the field of view and thus the centroid positions of the mCherry and GFP fluorophores are shifted relatively to each other differently in each endocytic event. When we average the contribution of the different trajectories together, the color shift averages out without significantly affecting the estimate of the relative position between the two fluorophores (Figure 1—figure supplement 4). However, larger color shift between the trajectory pairs, leads to larger uncertainty in the relative positions of the proteins (Figure 1—figure supplement 4).

*2) This work is critically dependent on the accuracy of trajectory alignment. Unfortunately the procedure of two color alignment is very hard to follow. For example, in the Materials and methods: “These trajectory pairs were then used to align the average trajectory of the protein of interest to the average trajectory of Abp1-GFP […] that aligned the Abp1-GFP average trajectory to the Abp1-mCherry trajectory in the trajectory pair.” I am confused about which average trajectory aligns to which and how they are averaged. Were the tracks of Abp1-mCherry and of the GFP-proteins aligned independently of each other? I do not find it logical that a two-step alignment has higher accuracy than a single step alignment, against what the authors claim. All additional procedures have to increase uncertainty. In the case of doubling the number of measurements, the uncertainty decreases only by a factor square root of 2. Without a detailed description and controlled test it is impossible to judge the real accuracy of the proposed procedures. The correctness of error propagation is critical for the conclusions, but it is not reported in detail*.

We have rewritten a more detailed description of the two color alignment procedure (See Materials and methods: “Two color alignment procedure”). All uncertainties in the procedure are taken into account, propagated and represented together with the trajectories. We complemented the description of the two color alignment procedure with a detailed description of our error propagation (See Materials and methods – “Two color alignment procedure”).

The accuracy of the trajectory alignment was supported by experimental controls (Figure 1—figure supplement 2) and (Figure 3—figure supplement 1). We also added simulated tests of the accuracy of our alignment procedure that prove the reliability and accuracy of our method for the conclusions we are making (See Materials and methods: Simulation of the accuracy of the two colors alignment procedure; Simulation of the robustness to systematic shifts between the two channels; Simulation of the accuracy of the trajectory averaging; Figure 1—figure supplement 3, Figure 1—figure supplement 4 and Figure 1—figure supplement 5).

The two steps procedure is dictated by a practical consideration: The simultaneous acquisition setup gives dimmer and noisier signals and we need to use longer exposure times than when the proteins are imaged individually. By first imaging the proteins individually we can thus use a higher frame rate (better temporal resolution) and record a brighter signal (better localization precision), leading to higher precision in the description of the dynamic behavior of the proteins. The trajectories obtained by simultaneous acquisition of two proteins give lower precision of protein dynamics, but are however sufficient to align together the average trajectories obtained by the imaging of individual proteins. We removed the potentially misleading sentence stating that the two-step aligment has higher accuracy. We think it was anyway redundant as the two color alignment method and its implications are described in their dedicated section as well as in the Material and methods.

*3) The number of molecules was estimated by a two-step comparison of fluorescence of the kinetochore proteins Nuf2 and Cse4. Unfortunately, the authors did not estimate the increase of the uncertainty by the additional step, but just multiplied the uncertainty of the first step by a scale factor (see Methods “Quantification of the number of molecules”). As such, the accuracy of number of molecules is overstated*.

The quantification of the average number of molecules is not a two-step procedure. We use only Nuf2 to quantify the fluorescence intensity of all the endocytic proteins as it has a brighter signal than Cse4. We used Cse4 to re-estimate the number of Nuf2 molecules. This allowed us to compare our Cse4 and Nuf2 ratio to Joglekar’s results, as a validation for our algorithm, and gave us an estimate of the uncertainty in the number of Nuf2 molecules, quantified with our algorithm, that we propagated in our quantifications.

When we checked the error propagation procedure we realized that a more appropriate description of the errors associated with the median is the Median Absolute Deviation (MAD) scaled for asymptotically normal consistency, not the standard error of the mean that we used before. We also corrected the number of Cse4 molecules from 4.9 to 5, as reported in Lawrimore, 2012. These corrections do not change any of the conclusions of our work. We added the detailed description of the error propagation in the section of the Materials and methods “Quantification of the number of molecules” and “Calibration of the fluorescence intensity curve of the trajectories with the number of molecules”.

*4) In the chapter “Assembly of Rvs proteins at the neck of the invagination” the authors described fitting procedures and estimations that are supposed to be numerical. Unfortunately, they did not provide any quantitative data and fit uncertainty estimation, but only a cartoon on*
Figure 4*. The small*
Figure 4
*just gives a crude value 100+- (?) molecules in the scission neck. The curve on*
Figure 4
*raises the question of whether a fraction of Rvs has to be on the plasma membrane side of the scission neck. After cutting the neck the membranes with Rvs move to opposite directions, with the vesicular part moving towards the interior and the PM part moving retrogradely. One could expect that it will be reflected in an oscillatory behavior of centroid with mean position in the center of the neck and high level of uncertainty. Unfortunately, this process is below the time resolution of the authors' technique. The curve on*
Figure 4
*shows a fast shift toward the vesicle. The authors commented that Rvs persists 2 sec longer on the vesicle than the plasma membrane. But another possible explanation is that a larger fraction of Rvs localized above the scission plane, something that would be inconsistent with the cartoon on*
Figure 4.

All the quantitative data used for the estimation of the Rvs coverage are listed in the “Materials and methods: Estimation of the membrane area covered by Rvs” and in the Table 2, where we list all the data used for the plots, the positions of the proteins and their abundances. We added an estimate of the variability in our fitting procedure in Figure 4—figure supplement 1. For clarity, we added a reference to the Table 2 in each figure legend where average trajectories are shown.

The model that we present for Rvs dynamics is based on three simple assumptions: 1) All the Rvs molecules locate on the membrane; 2) Rvs molecules are homogeneously distributed; 3) The density of Rvs molecules is the same as reported for the Rvs homolog endophilin in vitro. It is possible that a small fraction of Rvs molecules remains at the plasma membrane transiently after scission, but we would not have the resolution to resolve this possibility. The mechanism of scission indeed remains an important question, but a description of this very fast process at the level of individual molecules will require further method development.

*The spreads of red and green points on*
Figure 6
*are clearly different, but the difference in SD is rather minor (15 and 18nm)*.

We recalculated the SD estimate and it is correct. The standard deviation of the positions of patches, recovering after photobleaching, is 15.27 nm for Arc18 and 18.14 nm for Act1. The average positions are however different: 41 nm for Arc18-GFP and 27 nm for Act1. And their difference is significant: p value = 0.02.

Figure 1—figure supplement 2*: units of Y-axis are neither on the figure nor in the legend*.

We thank the reviewer for pointing out this error. The unit of the Y-axis are seconds and are now corrected in the plot.

Figure 1—figure supplement 2
*has 3 mistakes*:

*1) The optical slice depth 400nm is an overestimation. A well aligned confocal microscope has f ∼ 750nm*.

There are different definitions for the depth of field, which lead to different values. However, as shown in the plot (Figure 1—figure supplement 2) the underestimate of the invagination length would be marginal (2%) even with a depth of field of 1 μm. We rounded up the depth of field estimate in the figure legend to 500 nm.

*2) I suppose that the radius of a cell was taken equal 2.5 um, but it was not mentioned on the figure and in the legend*.

We now added it to the figure legend.

*3) In the formula on 2D there is mistake: the square root is missing. It has to be sqrt(1-f^2/(4 * r ^2)) * I. The estimation of maximal error is done with an over-optimistic optical depth and by a wrong formula that result in a 0.6% value in the legend*.

We thank the reviewer for pointing out this error. We corrected the formula. Fortunately, the missing square root in the formula overestimated our error. With the correct formula we get 0.4% with a depth of field of 450 nm, 0.5% with a depth of field of 500 nm, 1.1% with depth of field of 750 nm and 2% with a depth of field of 1 μm. We changed the figure legend accordingly.

*Moreover, this value is inconsistent with the value in the main text where it was given as 10%*.

The 10% of maximal underestimate that we report in the text takes into account both the variability of the angles of the trajectory movement in respect to the plasma membrane position (Figure 1—figure supplement 2), which would cause a projection artifact, and the projection due to the spherical geometry of the cell and the depth of field (Figure 1—figure supplement 2). See the main text Results section paragraphs seven and eight.

*On*
Figure 5
*the color scheme is far from optimal. I could not figure out from the figure and legend which tracks belong to Abp1 and which to Myo5*.

We improved the color scheme by darkening the color of Abp1. For additional clarity we made the plot in Figure 5 taller.

Video 2
*is packed with some uncommon codec, so I could not see it on two computers under 64bit Win7 and 32bit Win8 OS respectively*.

We are sorry for the trouble. We remade the movie and tested it under Mac OSX 10.9.5, Mac OSX 10.5.8, with QuickTime and VLC, and Windows XP, with QuickTime Player 7 and VLC, and it works fine now.

Reviewer #2

*Major novel findings are limited as no new concepts are discovered. The idea that actin polymerizes primarily at the base of the pits is not really new (due in part to work of Kaksonen himself; a google search for images of actin at yeast actin patches or clathrin coated pits reveals that most schematic cartoon place actin polymerization at the base of the invagination), although this is proved elegantly here. On the other hand, this study is highly quantitative, fluorescent proteins are expressed at endogenous levels and the findings are supported by a large number of controls*.

We strongly disagree with the lack of novelty in our work. First of all, the method we developed is conceptually novel and provides quantitative data about the endocytic machinery that was simply not accessible before. Furthermore, we describe the organization of coat proteins that are critical for normal vesicle budding, and provide a dynamic quantitative model of the membrane coverage by a BAR domain protein in vivo. We believe these data are essential for understanding the process of endocytic vesicle budding.

Different models for actin have indeed been proposed but there is no consensus in the scientific literature about the correct model. The available data has been indirect or has been derived from mutant cells whose normal endocytic process was impaired. Therefore, the question where actin primarily polymerizes has remained unsolved. Here we provide, for the first time, direct data to answer that question. We believe this is very important for understanding the mechanism of actin driven endocytosis.

*Issues to be addressed with discussion or new experimentation*:

*A major open question in the field is whether the dynamin homologue, Vps1, participates in fission at yeast actin patches. The authors mention Vps1, but do not show any data. Given the focus of this study, it would be great if the authors could analyze Vps1 and make a conclusive statement on this highly controversial topic*.

As suggested by the reviewer, we have analyzed the localization of Vps1. We have not been able to detect any fluorescently tagged Vps1 protein at the endocytic sites. We have imaged the Vps1-GFP also with TIRF microscopy, which gives very high sensitivity, but have failed to detect it at endocytic sites. We have expressed Vps1-GFP either alone or together with untagged Vps1, but have in all cases failed to detect any endocytic localization.

We conclude that if Vps1 is directly involved in endocytosis it must be present in a very low copy number, which is below the detection threshold. Unfortunately, as we cannot use our tracking method to study Vps1’s endocytic role, we think this question is outside of the scope of this current manuscript.

*The experiments with photobleaching of actin do not go past the time of scission (time = 0s). Thus, it remains unclear whether actin polymerization continues after scission. In this connection, several studies carried out on mammalian cells have demonstrated the occurrence of actin tails propelling newly formed endocytic vesicles. While tails were first described at sites of non-clathrin-dependent endocytosis (Merriefield and Almers, PMID: 10559868), a recent eLife paper demonstrated that also clathrin coated vesicles nucleate actin tails when the dephosphorylation of PI(4,5)P2 is impaired (Nandez et al., PMID: 25107275). Thus, while the present study proves what had been implicit in previous studies, i.e. that actin nucleation occurs at the base of the invagination, could some actin eventually nucleate also from the newly formed vesicle (for example at the site of the scar left by the fission)? The authors show that a pool of Rvs remains partially associated with the vesicle and one wonders whether some actin nucleating factors may be present at the same site*.

The yeast actin patches disassemble in few seconds after scission, so the possible time window for post scission actin polymerization is very short. We actually did perform photobleachings also later than the scission time, during the disassembly of the actin patch. After these late bleachings we did not observe any recovery, which would suggest that actin polymerization does not continue after scission. We don’t discuss these results in the manuscript because without a second marker to follow the vesicle, we cannot definitively know that the bleached vesicle is still in the focal plane. (This would be technically extremely challenging). A recent study in fission yeast showed that the movement of the newly formed vesicles just after scission is diffusive, and therefore not likely to be directed by an actin driven mechanism (4).

Different locations for the nucleation and polymerization of the actin filaments at the endocytic sites have been proposed in many previous studies but no conclusive evidence has been provided so far and the debate is still ongoing. A recent review by Mooren et al., summarizes this debate well (2012, Annu. Rev. Biochem; PMID: 22663081). Our results provide the first rigorous test of the different hypotheses, by direct visualization of the region of actin polymerization.

*Concerning Rvs (the endophilin homologue): it was shown that endophilin deficiency at neuronal synapses results in a defect in uncoating, most likely due to a deficient recruitment of synaptojanin, a PI(4,5)P2 phosphatase (Milosevic et al., PMID: 22099461). The Rvs pool that remains associated with the endocytic vesicles may be needed for uncoating, as Rvs167 binds a yeast synaptojanin. The authors may want to comment on this*.

This is an interesting possibility and we now discuss it in the text.

*The authors state that as the number of Las17 molecules start decreasing before fission, suggesting that fission may not be assisted by the force generated by actin polymerization. Yet, one cannot exclude that the remaining Las17 may still be needed to generate a force. Photobleaching experiments show continued actin nucleation at least until the time of fission. As tagged Las17 does not behave as wild type Las17, these experiments must be interpreted with caution*.

We proposed that nucleation of actin filaments decreases before scission happens. However, we agree that the polymerization of actin filaments may still continue and play a role in generating the force for scission. We changed the text to be more precise, also in agreement with the comments of reviewer 3 (please see the beginning of the subsection headed “The functional organization of the endocytic machinery”).

*Why is Las17 not drawn bound to the membrane in the cartoon of*
Figure 7*? Las17/nWASP are actin nucleators at the plasma membrane*.

The figure summarizes our tracking results by showing the protein symbols positioned according to their centroid positions in relation to the membrane invagination. To keep the illustration simple enough to be readable we cannot illustrate the known protein-protein or protein-membrane interactions. Furthermore, illustrating those interactions would also imply that we know when they all take place during endocytosis, which is not the case. We have clarified the legend of the Figure 7.

*The authors state that the number of Sla2 molecules remains relatively constant prior to scission, yet this is contrary to the data provided (*Figures 2, 3 and 7*), which shows that the number of Sla2 molecules starts declining a few seconds before scission*.

We have corrected our statement, as suggested.

*Despite the thorough analysis of the correlation between changes in membrane shape and the recruitment of endocytic components, it remains unclear what causes the initial buckling of the plasma membrane. This work favors the idea that the initial deformation coincides with the nucleation of actin (*Figure 7*), in line with the view presented in Kukulski, 2012. The stronger evidence in favor of this model is that in the presence of Latrunculin A, no invaginations are observed (Kukulski, 2012).*
[30]
*however, shows that there is an initial deformation of the membrane even in the presence of Latrunculin A. An alternative model suggests that early endocytic proteins like Syp1, Ede1 and Ent1 may act as the initiators of membrane curvature. What does the recruitment of these early endocytic proteins look like in this model and how does it relate to changes in membrane shape? Is the initial deformation the result of early endocytic proteins like Syp1, Ede1 and Ent1 coupled with actin polymerization? I note that, as often pointed out by Tom Kirchhausen who has extensively studied the structure of the clathrin lattice, it is unlikely that a flat clathrin lattice may convert to a curved clathrin lattice without undergoing a complete disassembly. Thus, it would seem more likely that the initial coated patch may already be curved. I realize that a thorough analysis of the spatio-temporal localization of early endocytic factors may go beyond the goal of this study, but these considerations should be discussed*.

With our tracking method, we could not detect any inward movement of the coat proteins that would indicate membrane bending prior to the appearance of the actin filaments. Sla2-GFP and GFP-Sla2, which are our best markers for the invagination dynamics among the early arriving endocytic proteins, show detectable movement only after actin locates at the endocytic site, in agreement with the data from [40] (Figure 2). [30] suggest that the membrane bending starts after the clathrin coat has assembled at the time of Vrp1 arrival (∼20 s before scission). Our coat protein trajectories do not show any inward movement during this time. Of course, a very shallow membrane bending might not be detectable by our method. All the other endocytic proteins that arrive before Sla2 (e.g. Syp1 and Ede1) can be knocked out in yeast without affecting the final vesicle budding process so they are unlikely to provide functionally significant membrane curvature. We added a discussion about this topic.

Reviewer #3

*This is rigorous, technically demanding work on an interesting biological process. I am generally enthusiastic about the work and the presentation, but think that the authors should have taken a broader perspective on their work, since their work confirms many of findings and ideas of others working on fission yeast. That previous work does not detract from the value of the present work, but it should not be ignored*.

We thank the reviewer for their enthusiasm about our work. We admit that our manuscript was unnecessarily narrowly focused on budding yeast results. We have extended the discussion of the relevant fission yeast literature as detailed below.

*Scientific issues*:

*Results section: “We thus produced a dataset in which all average trajectories are aligned to Abp1.” I am not concerned that some information was lost during averaging. The individual tracks in*
Figure 1
*average out all of the lateral motion of the patches, because the motions to the right and left of the normal (zero on the x-axis) to the plasma membrane cancel out. This representation may not give the casual reader the right impression about the patch motions, which would be more apparent with plots of the averages of the absolute values of the left and right positions*.

The average of the absolute values of the left and right positions is illustrated by Figure 1 (left panel). There, we plot the average trajectory oriented so that the average position along the plasma membrane, which is the movement right and left of the normal, is illustrated by the X-axis while the trajectory movement along the invagination direction is illustrated by the Y-axis. We clarified this point in the figure legend.

*In the first paragraphs of the Results section: It might be informative to compare the patch alignment method used here with that of Berro (MBoC, 2014). My impression is that the two methods work about equally well. Berro discusses the biological variability. Do you have any measure of the biological variability in your system, since you seem to assume none. You should state this assumption, if you made it*.

We did not assume that there is no biological variability. We cannot completely discern between the biological variability of the samples and the variability due to the imaging method. However, the variability in the dynamics and in the life times of the individual endocytic proteins that we observed (Figure 1—figure supplement 1) confirms that the endocytic process in yeast is extremely stereotypical. We added a sentence to the manuscript where we compare our method with the method of Berro.

*In the subsection “Organization of coat associated proteins”*, *in Results: An average localization precision of approximately 10 nm is extremely good; what is the evidence that the resolution is actually 10 nm?*

A localization precision of 10 nm or better is regularly achieved with the fluorophore Alexa Fluor 647 under comparable experimental conditions (Dempsey et al., “Evaluation of fluorophores for optimal performance in localization-based super-resolution imaging”, Nat Meth 8, 1027–1036, 2011, and our own work: Ries et al., “A simple, versatile method for GFP-based super-resolution microscopy via nanobodies,” Nat Methods 9, 582–584, 2012). This corresponds to a resolution (FWHM of the distribution of localizations around the true position of the fluorophore) of ∼25nm.

*In the beginning of the subsection “Assembly of Rvs proteins at the neck of the invagination”, in Results: The approach to calculating the membrane area covered by the Rvs dimers is identical to Arasada (2011) in fission yeast. Comparing the results may be informative*.

A comparison of the similarities and differences of the BAR domain protein behaviors between *S. cerevisiae* and *S. pombe* would indeed be very interesting. However, we feel that at this point a detailed comparison is difficult as the used methods are slightly different and the proteins studied in our manuscript and in Arasada paper are not the direct homologs. We added a citation to the Arasada paper for the approach.

*In the same section: Berro, MBoC (2014) has evidence that the movements of the actin-covered vesicle are diffusive in the cytoplasm*.

This is an excellent point. We have added a citation to Berro’s work to support the statement about the free diffusion of the vesicle in the cytoplasm.

*In the beginning of the subsection “Assembly of the actin cytoskeleton”: “The Myo5 trajectory remained almost stationary during the invagination of the plasma membrane, whereas the Las17 trajectory moved inward, but much less than Abp1, Arc18 and Act1 trajectories.” The behavior is similar in fission yeast (Arasada, 2011)*.

According to Arasada, the fission yeast Las17 moves much further at the tip of the invagination, while the budding yeast Las17 remains still closer to the base of the invagination. This is a very interesting difference between the species and could reveal interesting insight into the evolvability of the actin system. However, we feel that a quantitative side-by-side comparison between the two species will be needed.

*In the same section of the text, you state: “Note that because GFP-Act1 was expressed in addition to untagged actin, we do not know the ratio between tagged and untagged molecules that are recruited to the endocytic site and we can only provide relative abundance estimates.” Rather than using arbitrary units for Act1, measure the ratio of GFP-Act1 and wild type Act1with an anti-actin immunoblot and give the numbers of GFP-Act1 and the fraction tagged, allowing for a calculation of the total actin. Wu (2005) and Sirotkin (2010) explain how to do this*.

We followed the reviewer’s suggestion and quantified the amount of GFP-Act1 present at the endocytic patches. We then estimated by Western blotting the fraction of GFP-Act1 to be 9% ± 1% of the total cellular actin. We could thus estimate that the total peak number of actin molecules in actin patches is ∼3000. Assuming ∼200 filaments (based on the number of Arp2/3 molecules), each filament would be in average ∼40 nm long. We added this information to the manuscript: Figure 5, as well as “Materials and methods: Quantification of the ratio between GFP-Actin and the endogenous actin”.

*In the beginning of the subsection “Actin polymerizes at the base of the invagination”, in Results: The FRAP experiments are elegant and informative. Abp1 is a nice control. However the authors do not include the most obvious model in*
Figure 6*, namely that new branches are formed on pre-existing filaments in proportion to the local concentration of the NPFs and then grow in random directions. This is what is expected from the mechanism of branch formation and does not presuppose a mechanism to align the branches in a certain direction (which seems unlikely). Arasada (2011) uses this assumption and the data in*
Figure 6
*strongly support it. My impression is that the FRAP data are consistent with this model with local nucleation and random elongation, which the authors could simulate knowing the local concentrations of the two NPFs as established in this paper. Capping may be favored near the membrane (as assumed by some models of leading edge motility) and may bias the direction of growth, but the authors do not consider capping or other mechanisms to bias growth*.

We do not propose that actin branches are aligned in a certain direction and we agree that actin interactors, such as capping proteins, may bias the direction of growth, as commented in the Discussion. However, we still prefer a symbolic representation for Figure 6 as we think it makes it easier for the readers to understand how photobleaching would affect the centroid position of fluorescent actin patches in different models. To emphasize the symbolic nature of the actin representations we modified the figure legends of Figure 6 and Figure 6.

*In the subsection headed “The endocytic coat”, in the Discussion, you state: “The disassembly of Sla1 and End3 begins after membrane invagination has started, but several seconds before scission, suggesting that these proteins may be important during early stages of vesicle budding.” This supports the model of Chen (2013) with adaptor proteins binding actin filaments to provide mother filaments to start the branching process*.

We agree with the reviewer and have added a comment relating Chen’s model to our Results section.

*Still in the Discussion, the first paragraph of the subheading “Endocytic actin network”: Berro (2010) uses data similar to that in this paper to calculate the evolution of the rates of the main actin assembly reactions from the beginning to the end of the patch life, so one does not need to speculate about “these data suggest that the nucleation of new actin filaments stops before vesicle scission occurs.” Dissociation of the NPF's is enough to explain the slowing of the rate of branching*.

Here we want to specifically highlight our finding that actin filament nucleation decreases already before scission. This conclusion depends on our ability to time the scission very precisely, which has not been possible before.

We refer to Berro’s data when we discuss the overall similarity between *pombe* and *cerevisiae* results about actin patch assembly/disassembly.

*Subsection headed “The functional organization of the endocytic machinery”:*
Figure 7
*is the budding yeast version of fission yeast data in*
Figure 7
*and*
Table 1
*in*
[63]*. Arasada 2011 and Chen 2013 have additional quantitative data on other fission yeast actin patch proteins. The timing and numbers of molecules are remarkably similar in the two yeast in spite of >400 my of divergent evolution. This certainly deserves some discussion. The new data in this paper leads to conclusions similar to those in the fission yeast papers. Given this new data from budding yeast, it would be interesting to make a formal comparison with fission yeast actin patches, including the peak times, peak numbers and depth of penetration of each homologous molecule: actin, Arp2/3 complex, Wsp1 (Las17), Myo1 (Myo5), Cdc15 and Bzz1 (Rvs167) and End4 (Sla2)*.

We agree that the pioneering work from the Pollard lab on measuring the absolute numbers of molecules should be discussed in comparison to our data. We have added a discussion in text.

We do want to note that in addition to quantifying the assembly dynamics and the numbers of proteins, our work reveals how the different endocytic proteins are located in relation to each other and in relation to the evolving membrane shape at nano-scale resolution. This has never been done before.

We also agree that a comparison between the two species would be extremely interesting, but we feel that this would be better done in a separate paper due to the extensive amount of quantitative data available for both species (including biochemistry and genetics).

Figure 7
*legend: I would call the GFP-actin signal “polymerized actin” rather the “actin cytoskeleton,” since you use the latter term for actin filaments plus associated proteins*.

We agree and have changed “actin cytoskeleton” to “polymerized actin” (Figure 7).

*Since you can calculate the total numbers of polymerized actin molecules at each point along these curves and since you know the numbers of Arp2/3 complex, you should be able to make some simple calculations such as the lengths of the filaments and numbers of branches formed each second*.

We have added these estimates to the text as suggested.

Figure 7
*legend: “Time ≈ -8 s: actin polymerization starts.” This drawing is misleading, since it shows Sla2 binding a branch, whereas the biochemical evidence strongly shows that branching only occurs on the side of a pre-existing filament. Where do the pre-existing filaments come from? Chen (2013) has one possible source. The drawings of the later time points show that the network propagates correctly from mother filament to branch, but this is not obvious, so it would be helpful to have some clue showing that the first branches end up at the tip of the invagination*.

*The model considers the branches to be stable, but actually the branches and the filaments are actually turning over rapidly on the time scale of this figure (as shown by the simulations of Berro, 2010). It would be helpful if the figure or the text could indicate this turnover*.

*Note that Berro, MBoC, 2014, has more realistic scale drawings the filaments in slices through actin patches. If the actin numbers are similar in budding yeast actin patches, you could make less schematic drawing of the actin network base on his calculations*.

The cartoons in Figure 7 summarize our protein tracking data in the context of the membrane shapes. We use a simplified “dendritic actin network” illustration to symbolize actin filaments, but we do not want to imply that the branches are static or that they are precisely oriented. We considered more realistic representations, but felt that adding more data into this already busy figure is not helpful. We changed the figure legend to emphasize the intent of Figure 7 and to avoid false interpretations.

We comment Chen (2013) results in the main text.

[Editors' note: further revisions were requested prior to acceptance, as described below.]

*The referees were pleased that you took the reviewers' suggestions seriously. Nevertheless, Thomas Pollard (peer review) wrote: “My main concern with the original paper was its narrow point of view. The presentation generally ignored highly relevant work on other organisms, especially fission yeast. The new work on budding yeast offers valuable, new insights, but much of the story is remarkably similar to what was found in fission yeast. Making this clear highlights the amazing conservation of the mechanisms over the hundreds of millions of years since these two yeast diverged. These comparisons detract in no way from the authors' accomplishments. Also, the authors seem to have misunderstood several suggestions to compare the data from the two yeasts. They declined to do so, saying that this would require new experiments. However, all they needed to do was to plot the published data on the same graphs as their new data and comment on what is the same and what is different*.

We followed Pollard’s suggestion and we carefully compared our data with the quantitative data available for *S. pombe.* We now discuss the similarities and the differences between the two species in a new paragraph (second paragraph of the subsection “The functional organization of the endocytic machinery”).

*Second, citations: The authors added citations to some of the work on fission yeast, but missed some opportunities to inform a reader about what was known. Here are two examples*:

*Counting total actin from the numbers of GFP-actin and the ratio of GFP-actin to untagged actin: the new data look good, but the method developed by Wu was not cited*.

We added the citation to Wu’s work both in the discussion of our Results and in the Materials and methods section, “Quantification of the ratio between GFP-Act1 and the total actin”.

*Idea that adaptor proteins bind actin filaments to provide mother filaments for Arp2/3 complex to start the branching process: The revised text makes this point and cites four papers, three dealing with the discovery of the adapter proteins, none of which mention Arp2/3 complex. This gives the impression that these early paper rather than the Chen paper came up with the idea*.

We changed the paragraph so that the contribution of the different citations is clear. In particular, we give credit to Chen’s paper when describing the importance of the mother filaments to start the branching process.

*Finally, (not essential) in terms of the idea that the dissociation of the NPF's is enough to explain the slowing of the rate of branching: I was hoping that they would use their data to calculate the rates of the reactions, as done before by others. This would be nice but is not essential*.

It will indeed be interesting to model the reaction rates of actin polymerization, but we feel that is a bit out of the scope of the current manuscript, where our goal was to visualize the assembly dynamics of the whole endocytic machinery.

In addition, we corrected the label on the right y-axis on Figure 7.